# DeCCaF: Deferral under Cost and Capacity Constraints Framework

## Abstract

The *learning to defer* (L2D) framework aims to improve human-AI collaboration systems by deferring decisions to humans when they are more likely to make the correct judgment than a ML classifier. Existing research in L2D overlooks key aspects of real-world systems that impede its practical adoption, such as: i) neglecting cost-sensitive scenarios; ii) requiring concurrent human predictions for every instance of the dataset in training and iii) not dealing with human capacity constraints. To address these issues, we propose the *deferral under cost and capacity constraint framework* (DeCCaF). A novel L2D approach: DeCCaF employs supervised learning to model the probability of human error with less restrictive data requirements (only one expert prediction per instance), and uses constraint programming to globally minimize error cost subject to capacity constraints. We employ DeCCaF in a cost-sensitive fraud detection setting with a team of 50 synthetic fraud analysts, subject to a wide array of realistic human work capacity constraints, showing that DeCCaF significantly outperforms L2D baselines, reducing average misclassification costs by 9 %. Our code and testbed are available at https://anonymous.4open.science/r/deccaf-1245

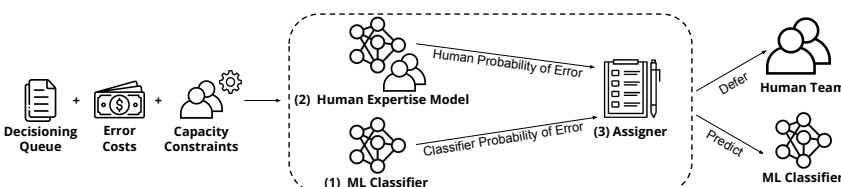

Figure 1: Schematic Representation of DeCCaF

## 1 Introduction

An increasing body of recent research has been dedicated to human-AI collaboration (HAIC), with several authors arguing that humans have complementary sets of strengths and weaknesses to those of AI (De-Arteaga et al., 2020; Dellermann et al., 2019). Collaborative systems have demonstrated that humans are able to rectify model predictions in specific instances (De-Arteaga et al., 2020), and have shown that humans collaborating with an ML model may achieve synergistic performance: a higher performance than humans or models alone (Inkpen et al., 2022). In high stakes scenarios where ML models can outperform humans, such as healthcare (Gulshan et al., 2016), HAIC systems can help address concerns regarding the safety of ML model use (*i.e.* the effect of changes in the data distribution (Gama et al., 2014)), by ensuring humans are involved in the decision making process.

The state-of-the-art framework to manage assignments in human-AI collaboration is *learning to defer* (L2D) (Charusaie et al., 2022; Hemmer et al., 2022; Raghu et al., 2019b;a; Mozannar & Sontag, 2020; Mozannar et al., 2023; Madras et al., 2018). L2D aims to improve upon previous approaches such as rejection learning (Chow, 1970; Cortes et al., 2016), which defer based solely on the ML model's confidence, by also estimating the human confidence in a given prediction and passing the instance to the decision-maker who is most likely to make the correct decision.

Previous work in L2D does not address several key aspects of collaborative systems. Training these systems requires large amounts of data on human decisions. While some require multiple human predictions per instance (Raghu et al., 2019b;a), most require human predictions to exist for every single training instance (Madras et al., 2018; Mozannar & Sontag, 2020; Verma & Nalisnick, 2022; Hemmer et al., 2022). In real-world applications, we often have limited data availability (De-Arteaga et al., 2020; Gulshan et al., 2016) due to limitation in human work capacity (*i.e.* it is not feasible to have every expert review every instance). Current L2D methods also neglect human capacity limitations when deferring. While some authors propose including a cost of deferral to moderate the amount of cases that can be passed on to humans (Mozannar & Sontag, 2020), the individual work capacity of each human is not considered. Should there be an expert that consistently outperforms the model and all other experts, the optimal assignment would be to defer every case to that expert, which in practice is not feasible. Furthermore, current work neglects cost-sensitive scenarios, where the cost of misclassification can be class or even instance-dependent (e.g. in medicine, false alarms are generally considered less harmful than failing to diagnose a disease).

To address the aforementioned L2D limitations, we propose the *deferral under cost and capacity constraints framework* (DeCCaF), a novel method to manage assignments in cost-sensitive human-AI decision-making while respecting human capacity constraints. Our method is comprised of three separate parts, represented schematically in figure 1: (1) an ML classifier modelling the probability of the target class given the instance features; (2) a human expertise model, which models the error probabilities of each of the experts within a team; (3) an assigner, which computes the best possible set of assignments given misclassification costs and capacity constraints. Due to the lack of sizeable datasets with multiple humans' predictions, and the high costs associated with producing one, we empirically validate our method in a realistic cost-sensitive fraud detection setting, where a team of 50 instance dependent synthetic fraud experts are tasked with reviewing an ML Model's alerts. We conclude that DeCCaF outperforms L2D baselines in this scenario, resulting in a decrease of 9 % in the average misclassification cost. To summarize, our contributions are the following:

- DeCCaF, a novel L2D method that models human behavior under limited data availability, using constraint programming to calculate the optimal set of assignments.
- A novel benchmark of complex, feature-dependent synthetic expert decisions, generated by applying instance-dependent label noise.
- Experimental evidence that DeCCaF outperforms the baselines in an extensive set of capacity-constraint scenarios in a realistic, cost-sensitive fraud detection task.

## 2 RELATED WORK

### 2.1 CURRENT L2D METHODS

The simplest deferral approach in the literature is *rejection learning* (ReL) (Chow, 1970; Cortes et al., 2016). In a human-AI collaboration setting, ReL defers to humans the instances that the model rejects to predict. (Madras et al., 2018; Raghu et al., 2019a). A simple example (Hendrycks & Gimpel, 2017) would be to produce uncertainty estimates for the model prediction in each instance, rejecting to predict if the uncertainty is above a given threshold.

Madras et al. (2018) criticize ReL, arguing that it does not consider the performance of the human involved in the task, and propose *learning to defer* (L2D), where the classifier and assignment system are trained jointly, taking into account a single model and a single human, and accounting for human error in the training loss. Many authors have since contributed to the single-expert framework (Mozannar & Sontag, 2020; Mozannar et al., 2023). Mozannar & Sontag (2020) show that the loss proposed by Madras et al. (2018) is inconsistent, proposing a consistent surrogate loss that yields better results in testing. Verma & Nalisnick (2022) formulate a critique to the approach of Mozannar & Sontag (2020), demonstrating that their surrogate loss has a degenerate parameterization, causing miscalibration of the estimated probability of correctness of the expert. Keswani et al. (2021) observe that decisions can often be deferred to one or more humans out of a team, expanding L2D to the multi-expert setting. Verma et al. (2023) propose a new consistent and calibrated surrogate loss, generalizing the work of Verma & Nalisnick (2022) to the multi-expert setting. That approach outperforms a multi-expert adaptation of the surrogate loss of Mozannar & Sontag (2020), due to the same calibration problems observed in the single expert setting. All aforementioned studies focus

on deriving surrogates for the $0 - 1$ loss, meaning they are not directly applicable to cost-sensitive scenarios, where the cost of erring can be instance dependent.

Another facet of the problem is the interplay between the two components in a L2D system. Mozannar & Sontag (2020) argue that the main classifier should be allowed to specialize on the instances that will be assigned to it, in detriment of those that will not. This is done by jointly training the classifier and the rejector, and choosing not to penalize the classifier's mistakes on instances that the rejector chooses to defer to the expert. While this approach may enable the system to achieve an improvement in overall performance, it may not be suitable for real world applications. By design, the instances that are most likely to be deferred are those in which the classifier will perform worse. This would result in a classifier that is not suitable for domains where the AI score advises the human decision-makers, as the ML model will have unpredictable behaviour in the deferred instances. Furthermore, specialization may make the system highly susceptible to changes in human availability, should the AI have to predict on instances that were previously meant to be deferred.

The key drawback of the aforementioned approaches is that they require predictions from every human team member, for every training instance. To train these systems, practitioners must incur in further costs gathering a set of every human's prediction for every instance, as in real-world applications, such amounts of data are rarely available. In most cases, due to the limited human work capacity, each instance is reviewed by a single human worker (De-Arteaga et al., 2020) or by a small fraction of a larger team (Gulshan et al., 2016). We should then consider algorithms that can be developed under limited data availability, that is, considering that each instance will only have a single expert prediction associated with it.

When deferring, human work capacity constraints are rarely considered in multi-expert L2D, where the goal is to find the best decision-maker for each instance, disregarding the amount of instances that are deferred to each individual agent. While some L2D methods allow the user to set a deferral cost to control the amount of deferrals (Mozannar & Sontag, 2020), the limited work capacity of each individual team member often goes unaddressed. In our work, we propose a L2D algorithm that can be utilized in cost sensitive scenarios, trained with restrictive data requirements, and employed under individual human work-capacity constraints.

## 2.2 SIMULATION OF HUMAN EXPERTS

Due to the lack of sizeable, public, real-world datasets with multiple experts, most authors use a *label noise* approach to produce arbitrarily accurate expert predictions on top of established datasets in ML literature. Mozannar & Sontag (2020) use CIFAR-10 (Krizhevsky et al., 2009), where they simulate an expert with perfect accuracy on a fraction of the 10 classes, but random accuracy on the others (see also the work by Verma & Nalisnick (2022) and Charusaie et al. (2022)). The main drawback of these synthetic experts is that their expertise is rather simplistic, being either feature-independent or only dependent on a single feature or concept. This type of approach has been criticised by authors such as Zhu et al. (2021) and Berthon et al. (2021), who argue that *instance-dependent label noise* (IDN) is more realistic, as human errors are likely to be dependent on the difficulty of a given task, and, as such, should dependend on its features. In this work, we propose an IDN approach to simulate more complex and realistic synthetic experts.

## 3 METHOD - DeCCaF

To overcome the limitations that make L2D unfeasible to apply in real-world settings, we propose DeCCaF, a novel approach to manage assignments in the multi-expert setting, while respecting human capacity constraints and limited data availability. Over the following sections we will formalize our definition of capacity constraints, describe the training process of our method, and discuss cost minimization algorithms under capacity constraints.

### 3.1 DEFINITION OF CAPACITY CONSTRAINTS

Firstly, we formalize how to define capacity constraints. Humans are limited in the number of instances they may process in any given time period (e.g., work day). In real-world systems, human capacity must be applied over batches of instances, not over the whole dataset at once (e.g. balancing

the human workload over an entire month is not the same as balancing it daily). A real-world assignment system must then process instances taking into account the human limitations over a given "batch" of cases, corresponding to a pre-defined time period. We divide our dataset into several batches and, for each batch, define the maximum number of instances that can be processed by each expert. In any given dataset comprised of $N$ instances, capacity constraints can be represented by a vector $\boldsymbol{b}$, where component $b_i$ denotes which batch instance $i \in \{1, ..., N\}$ belongs to, as well as a human capacity matrix $H$, where element $H_{b,j}$ is a non-negative integer denoting the number of instances expert $j$ can process in batch $b$.

## 3.2 Data Requirements, ML Classifier, and Human Expertise Model

**Data and ML Model.** Assume that for each instance $i$ we have a ground truth label $y_i \in \mathcal{Y} = \{0, 1\}$, as well as a vector $\boldsymbol{x}_i \in \mathcal{X}$ representing its features as well as any known additional information the expert may have access to (*e.g.* ML Model score). Assume also that each instance has an associated cost of misclassification $c_i$, and a prediction $m_{i,j} \in \mathcal{Y}$ from a given expert $j$. Having access to a training set $S = \{\boldsymbol{x}_i, y_i, m_{i,j}, c_i\}_{i=1}^N \sim D$ with $j \in \{1, ..., N\}$, our aim is to estimate the $\mathbb{P}(m_{i,j} \neq y_i | \boldsymbol{x}_i, j)$, with $j \in \{1, ..., N\}$.

**ML Model.** We assume the ML classifier is trained to predict the label $y_i$. As long as its output is calibrated, we can use it as an estimate of its error probability.

**Human expertise model.** When data pertaining to each individual expert is limited, it may be beneficial to jointly model the entire team. To do so, we consider that our input space is $\mathcal{X} \cup \{1, ..., J\}$, where $j$ denotes the expert's id. Having access to the ground truth $y_i$, we can determine if the expert was correct and what type of error, if any, was committed on a given instance. In the binary case, we distinguish between false positives and false negatives, as these may incur different costs. As such, our human error probability prediction model is composed of a single classifier $h : \mathcal{X} \cup \{1, ..., J\} \rightarrow \mathcal{O}$, where $\mathcal{O} = \{\text{'fn'}, \text{'fp'}, \text{'tn'}, \text{'tp'}\}$ represents the outcome space.

The learning objective function is the multi-class softmax log-loss, a calibrated surrogate to the 0-1 loss (Lapin et al., 2017), in order to obtain well calibrated probability estimates for each outcome probability. In models which fit the statistical query model (Kearns, 1998), the expected value of the loss function $L$, $\mathbb{E}_{(\boldsymbol{x},y) \sim D}[L(\boldsymbol{x}, y)]$, is approximated as $\frac{1}{N} \sum_{(\boldsymbol{x}_i, y_i)} L(\boldsymbol{x}_i, y_i)$, assuming each of the training instances is equally important. However, as previously mentioned, instances may have different misclassification costs. Knowing that the data is sampled from a distribution $D$, and that each datum has an associated misclassification cost $c$, the goal is then to learn the classifier $h$ that minimizes the expected cost, $\mathbb{E}_{\boldsymbol{x},y,c \sim D}[c\mathbf{1}_{(\text{g}(\boldsymbol{x}) \neq \text{y})}]$. Minimizing a surrogate to the 0-1 loss ensures that we are minimizing the expected error rate $\mathbb{E}_{\boldsymbol{x},y,c \sim D}[\mathbf{1}_{(\text{g}(\boldsymbol{x}) \neq \text{y})}]$, which is misaligned with our objective. Zadrozny et al. (2003) show that if we have examples drawn from a different distribution

$$\tilde{D}(x, y, c) = \frac{c}{\mathbb{E}_{c \sim D}[c]} D(x, y, c), \text{ then } \mathbb{E}_{\boldsymbol{x},y,c \sim \tilde{D}}[\mathbf{1}_{(\text{g}(\boldsymbol{x}) \neq \text{y})}] = \frac{1}{\mathbb{E}_{c \sim D}[c]} E_{\boldsymbol{x},y,c \sim D}[c\mathbf{1}_{(\text{g}(\boldsymbol{x}) \neq \text{y})}] \quad (1)$$

This shows that selecting $h$ to minimize the error rate under $\tilde{D}$ is equivalent to selecting $h$ to minimize the expected misclassification cost under $D$. This means we can obtain our desired classifier by training it under the distribution $\tilde{D}$, using the softmax log-loss. To train a classifier under $\tilde{D}$, a common approach (Zadrozny et al., 2003; Elkan, 2001) is to re-weight the instances according to their costs. In this way, we obtain a classifier $h$ which prioritizes correctness according to the instance's weights. As the probability estimates obtained through minimizing the re-weighted expectations will be calibrated under $\tilde{D}$, we transform them so that they are calibrated w.r.t to $D$.

## 3.3 Minimization objective

Having the ability to model the error probabilities of the involved decision-makers, we can now combine them into an estimate of the misclassification cost, to be minimized across a set of instances. Having an estimate for the probability that expert $j$ will predict a false positive in instance $\hat{\mathbb{P}}(\text{FP})_{i,j} = \hat{\mathbb{P}}(m_{i,j} = 1 \wedge y_i = 0)$ and the probability of a false negative $\hat{\mathbb{P}}(\text{FN})_{i,j} = \hat{\mathbb{P}}(m_{i,j} = 0 \wedge y_i = 1)$, we can construct the predicted misclassification cost of deferring instance $i$ to expert $j$:

$$\hat{C}_{i,j} = \lambda \hat{\mathbb{P}}(\text{FP})_{i,j} + \hat{\mathbb{P}}(\text{FN})_{i,j}, \quad \lambda \in \mathbb{R}_+, \quad (2)$$

where $\lambda$ controls the relative weight of false positives and false negatives. This allows for optimization in cost-sensitive scenarios where errors have different costs. The objective of our algorithm is to minimize the expected cost over a set of assignments.

### 3.4 MINIMIZATION ALGORITHM

There are various possible approaches to minimizing the expected misclassification cost over a set of instances. A greedy approach selects the available expert with the lowest predicted cost for each instance. However, sequentially choosing the best decision-maker may not be the optimal choice. For example, if a decision-maker is universally better than the rest of the team, assigning instances sequentially would result in overusing the best decision-maker on the initial instances in a given batch. Ideally, they would decide on the cases where the other experts are most likely to err.

We express the assignment problem, in each batch $b$, as a minimization objective subject to a set of constraints. For each batch $b$ comprised of $n$ instances, consider the matrix $A$, were each element $A_{i,j}$ is a boolean variable that denotes if instance $i \in \{1, ..., n\}$ is deferred to expert $j \in \{1, ..., J\}$. The optimal assignment is given by:

$$A^* = \underset{A \in \{0,1\}^{n \times J}}{\arg\min} \sum_i \sum_j \hat{C}_{ij} A_{ij},$$

$$s.t. \quad \sum_i A_{ij} = H_{b,j}, \text{ for } j \in \{1, 2, ..., J\} \text{ and } \sum_j A_{ij} = 1, \text{ for } i \in \{1, 2, ..., n\}. \tag{3}$$

The first constraint refers to human decision capacity: the number of instances assigned to each decision-maker is predefined in the problem statement. This constraint may be changed to an inequality expressing the maximum number of assignments per decision-maker. The second constraint states that each instance can only be assigned to one decision-maker. We solve the assignment problem using the constraint programming solver CP-SAT from Google Research's OR-Tools.

## 4 EXPERIMENTAL SETTING

In this section we present our experimental setting. We start by proposing a novel method for generating synthetic expert prediction for tabular datasets, followed by a description of the generated synthetic expert team. We then describe our chosen classification task, specifying the realistic deferral conditions and human capacity constraints under which our system must operate.

### 4.1 SIMULATED EXPERTS

Our expert generation approach is based on *instance-dependent noise*, in order to obtain more realistic experts, whose probability of error varies with the properties of each instance. We generate synthetic predictions by flipping each label $y_i$ with probability $\mathbb{P}(m_{i,j} \neq y_i | \boldsymbol{x}_i, y_i)$. In some HAIC systems, the model score for a given instance may also be shown to the expert (Amarasinghe et al., 2022; De-Arteaga et al., 2020; Levy et al., 2021), so an expert's decision may also be dependent on an ML model score $m(\boldsymbol{x}_i)$. We define the expert's probabilities of error, for a given instance, as a function of its features, $\boldsymbol{x}_i$, and the model score $m(\boldsymbol{x}_i)$,

$$\begin{cases} \mathbb{P}(\hat{y}_i = 1 | y_i = 0, \boldsymbol{x}_i, M) = \sigma\left(\beta_0 - \alpha \frac{\boldsymbol{w} \cdot \boldsymbol{x}_i + w_M M(\boldsymbol{x}_i)}{\sqrt{\boldsymbol{w} \cdot \boldsymbol{w} + w_M^2}}\right) \\ \mathbb{P}(\hat{y}_i = 0 | y_i = 1, \boldsymbol{x}_i, M) = \sigma\left(\beta_1 + \alpha \frac{\boldsymbol{w} \cdot \boldsymbol{x}_i + w_M M(\boldsymbol{x}_i)}{\sqrt{\boldsymbol{w} \cdot \boldsymbol{w} + w_M^2}}\right), \end{cases} \quad M(\mathbf{x}_i) = \begin{cases} \frac{m(\boldsymbol{x}_i) - t}{2t}, & m \leq t \\ \frac{m(\boldsymbol{x}_i) - t}{2(1-t)}, & m > t, \end{cases}$$

where $\sigma$ denotes a sigmoid function and $M$ is a transformed version of the original model score $m \in [0, 1]$. Each expert's probabilities of the two types of error are parameterized by five parameters: $\beta_0, \beta_1, \alpha, \mathbf{w}$ and $w_M$. The weight vector $\mathbf{w}$ embodies a relation between the features and the probability of error. To impose a dependence on the model score, we can set $w_M \neq 0$. The feature weights are normalized so that we can separately control, with the $\alpha$ parameter, the overall magnitude of the variation of the probability of error due to the instance's features. The values of $\beta_1$ and $\beta_0$ control the base probability of error. The motivation for this approach is explained further in section B.1.

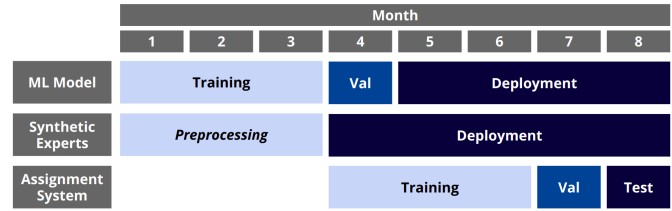

Figure 2: Chosen temporal splits for ML model training, development and testing of assignment systems.

To simulate a wide variety of human behavior, we created four distinct expert groups. The first is a *Standard* expert group: on average as unfair as the model, dependent on model score and, on average, twelve different features. The three other types of expert are variations on the *Standard* group. i) Unfair: experts which are more likely to incorrectly reject an older customer's application. ii) *Model agreeing*: experts which are heavily swayed by the model score. iii) *Sparse*: experts which are dependent on fewer features. The values for the parameters used to generate each expert are available in section C.2 of the Appendix.

## 4.2 DATASET

As the base dataset, we choose to use the publicly available bank-account-fraud tabular dataset (Jesus et al., 2022). This dataset is comprised of one million rows, with each instance representing a bank account opening application, having a label that denotes if the instance is a fraudulent application (1) or a legitimate one (0). The features of each instance contain information about the application and the applicant. The task of a decision maker (automated or human) is to either accept (predicted negative) or reject (predicted positive) it. A positive prediction results in a declined application. As such, false positives in account opening fraud can significantly affect a person's life (with no possibility to open a bank account or to access credit). This is thus a cost-sensitive problem, where the cost of a false positive must be weighed against the cost of a false negative. The optimization goal is to maximize recall at a fixed FPR rate (we use 5%), which implicitly establishes a relationship between the costs. This task also entails fairness concerns, as ML models trained on this dataset tend to raise more false fraud alerts (false positive errors) for older clients ($\geq 50$ years), thus reducing their access to a bank account.

## 4.3 HAIC SETUP

We choose temporal splits of the dataset that aim to emulate a realistic scenario as close as possible, represented in figure 2. To do so, we first train a fraud detection ML classifier. This model is trained on the first three months of the dataset and validated on the fourth month. We utilize the LightGBM (Ke et al., 2017) algorithm, due to its proven high performance on tabular data (Shwartz-Ziv & Armon, 2022; Borisov et al., 2022). Details on the training process of the classifier are given in Section C.1 of the Appendix. Our simulated experts are assumed to act alongside the ML model on the period ranging from the fourth to the eighth month. There are several possible ways for models and humans to cooperate. In L2D testing, it is often assumed any instance can be deferred to either the model or the expert team. However, in a real world setting, it is common to use an ML model to raise alerts that are then reviewed by human experts (De-Arteaga et al., 2020; Han et al., 2020). Without an assignment method, the decision system would function as follows: a batch of instances is processed by the model, a fraction of the highest scoring instances are flagged for human review, and, finally, these instances are randomly distributed throughout the humans in the decision team, who make the final decision. The rest of the instances are automatically accepted.

By assuming that the alert review system is employed from months four to seven, we can construct a dataset that would correspond to human predictions gathered over this period. Using this data, we train our assignment algorithms with the data of months four to six, validating them on the seventh month. Testing is done by creating a new deferral system, where the cases flagged by the ML classifier for review are distributed to the humans according to an assignment algorithm trained on

the gathered data. In this case, the assignment algorithm only has to choose which human to pass the instance to, as the ML classifier raises the alerts

## 4.4 CAPACITY CONSTRAINTS

Our capacity constraints are denoted by the batch vector and capacity matrix, as defined in Section 3.1. To define the batch vector, we have to define the number of cases in each batch, then distribute instances throughout the batches. Different batch sizes allow for testing assignment methods over long and short horizons. To define the capacity matrix, we consider 4 separate parameters. (1) Deferral_rate: maximum fraction of each batch that can be deferred to the human team; (2) Distribution *homogeneous* or *variable*. Should the distribution be *homogeneous*, every expert will have the same capacity; otherwise, each expert's capacity is sampled from $\mathcal{N}(\mu_d = \text{Deferral\_rate} \times \text{Batch\_Size}/N_{experts}, 0.2 \times \mu_d)$, chosen so that each expert's capacity fluctuates around the value corresponding to an homogeneous distribution; (3) Absence rate, defined as the fraction of experts that are absent in each batch. This allows for testing several different team configurations without generating new experts, or scenarios where not all experts work in the same time periods. (4) Expert Pool, defined as which types of experts (standard, unfair, sparse or model agreeing) can be part of the expert team.

To allow for extensive testing of assignment systems, we create a vast set of capacity constraints. In Table 1, under "Scenario Properties", we list the different combinations of settings used. For each combination, several seeds were set for the batch, expert absence, and capacity sampling.

## 5 RESULTS

In order to evaluate DeCCaF and ensure the robustness of results, both the OvA (One vs. All) method and human expertise model are trained in 5 distinct variants of the training set, created by varying the random distribution of cases throughout months 4 to 7. They are then tested under a set of 300 capacity constraints. The ML model yields a recall of 57.9% in validation, for a threshold of $t = 0.051$, chosen in validation to obtain 5% FPR. Additional statistics regarding the ML model's performance are detailed in Section C.1 of the Appendix.

### 5.1 BASELINES

**One vs. All** For a L2D baseline, we use the multi-expert learning to defer "One versus All" (OvA) algorithm, proposed by Verma et al. (2023). This method originally takes training samples of the form $D_i = \{\boldsymbol{x}_i, y_i, m_{i,1}, ..., m_{i,J}\}$. This method assumes the existence of a set of every expert's predictions for each training instance, however, this is not strictly necessary.

The OvA model relies on creating a classifier $h : \mathcal{X} \rightarrow \mathcal{Y}$ and a rejector $r : \mathcal{X} \rightarrow \{0, 1, ..., J\}$. When $r(\boldsymbol{x}_i) = 0$, the classifier makes the decision, and when $r(\boldsymbol{x}_i) = j$, the decision is deferred to the $j$th expert. The classifier is composed of K functions: $g_k : \mathcal{X} \rightarrow \mathbb{R}$ for $k \in \{1, ..., K\}$ where $k$ denotes the class index. These are related to the probability that an instance belongs to class $k$. The rejector, similarly, is composed of J functions: $g_{\perp,j} : \mathcal{X} \rightarrow \mathbb{R}$ for $j \in \{1, ..., J\}$, which are related to the probability that expert $j$ will make the correct decision regarding said instance. The authors propose combining the functions $g_1, ..., g_K, g_{\perp,1}, ..., g_{\perp,J}$ in an OvA surrogate for the 0-1 loss. The OvA multi-expert L2D surrogate is defined as:

$$\Psi_{\text{OvA}}(g_1, ..., g_K, g_{\perp,1}, ..., g_{\perp,J}; \boldsymbol{x}, y, m_1, ..., m_J) = \Phi[g_y(\boldsymbol{x})] + \sum_{y' \in \mathcal{Y}, y' \neq y} \Phi[g'_y(\boldsymbol{x})]$$

$$+ \sum_{j=1}^{J} \Phi[-g_{\perp,j}(\boldsymbol{x})] + \sum_{j=1}^{J} \mathbb{I}[m_j = y](\Phi[g_{\perp,j}(\boldsymbol{x})] - \Phi[-g_{\perp,j}(\boldsymbol{x})])$$

where $\Phi : \{\pm 1\} \times \mathbb{R} \rightarrow \mathbb{R}_+$ is a binary surrogate loss. Verma et al. (2023) then prove that the minimizer of the pointwise inner risk of this surrogate loss can be analyzed in terms of the pointwise minimizer of the risk for each of the $K + J$ underlying OvA binary classification problems, concluding that the minimizer of the pointwise inner $\Psi_{\text{OvA}}$-risk, $\boldsymbol{g}^*$, is comprised of the minimizer

Table 1: Baselines vs DeCCaF: Comparison of misclassification cost 5 and Predictive Equality (ratio of group-wise FPR). Customers with age over 50 years have larger FPR. Error bars denote 95% CI

| | | Scenario Properties | | | ReL | | OvA | | DeCCaF$_{greedy}$ | | DeCCaF$_{linear}$ | |
|---|---|---|---|---|---|---|---|---|---|---|---|---|
| Batch | $N_{exp}$ | Dist. | Def. | Pool | Cost | PE | Cost | PE | Cost | PE | Cost | PE |
| 1000 | 0 | homogenous | 0.2 | all | $806_{\pm19}$ | 0.19 | $800_{\pm35}$ | 0.17 | $733_{\pm39}$ | 0.22 | $\mathbf{725}_{\pm50}$ | 0.21 |
| 1000 | 0 | homogenous | 0.5 | all | $784_{\pm18}$ | 0.21 | $773_{\pm63}$ | 0.20 | $673_{\pm41}$ | 0.27 | $\mathbf{671}_{\pm41}$ | 0.27 |
| 1000 | 0 | variable | 0.2 | all | $806_{\pm40}$ | 0.17 | $795_{\pm39}$ | 0.16 | $728_{\pm39}$ | 0.21 | $\mathbf{724}_{\pm47}$ | 0.21 |
| 1000 | 0 | variable | 0.5 | all | $786_{\pm31}$ | 0.20 | $772_{\pm58}$ | 0.20 | $676_{\pm43}$ | 0.26 | $\mathbf{672}_{\pm40}$ | 0.27 |
| 1000 | 0.5 | homogenous | 0.2 | all | $814_{\pm44}$ | 0.18 | $804_{\pm34}$ | 0.16 | $743_{\pm42}$ | 0.21 | $\mathbf{737}_{\pm42}$ | 0.21 |
| 1000 | 0.5 | homogenous | 0.5 | all | $806_{\pm45}$ | 0.21 | $781_{\pm46}$ | 0.20 | $683_{\pm56}$ | 0.26 | $\mathbf{679}_{\pm48}$ | 0.27 |
| 1000 | 0.5 | variable | 0.2 | all | $812_{\pm33}$ | 0.18 | $802_{\pm36}$ | 0.16 | $741_{\pm43}$ | 0.21 | $\mathbf{738}_{\pm40}$ | 0.21 |
| 1000 | 0.5 | variable | 0.5 | all | $792_{\pm28}$ | 0.20 | $781_{\pm47}$ | 0.20 | $683_{\pm61}$ | 0.26 | $\mathbf{681}_{\pm51}$ | 0.27 |
| 5000 | 0 | homogenous | 0.2 | all | $793_{\pm42}$ | 0.19 | $807_{\pm32}$ | 0.16 | $\mathbf{732}_{\pm38}$ | 0.21 | $732_{\pm55}$ | 0.21 |
| 5000 | 0 | homogenous | 0.5 | all | $776_{\pm30}$ | 0.21 | $777_{\pm68}$ | 0.20 | $675_{\pm37}$ | 0.27 | $\mathbf{670}_{\pm26}$ | 0.27 |
| 5000 | 0 | variable | 0.2 | all | $802_{\pm28}$ | 0.19 | $806_{\pm31}$ | 0.16 | $\mathbf{732}_{\pm38}$ | 0.21 | $732_{\pm54}$ | 0.21 |
| 5000 | 0 | variable | 0.5 | all | $786_{\pm35}$ | 0.20 | $779_{\pm62}$ | 0.20 | $678_{\pm38}$ | 0.27 | $\mathbf{675}_{\pm31}$ | 0.27 |
| 5000 | 0.5 | homogenous | 0.2 | all | $820_{\pm25}$ | 0.19 | $809_{\pm37}$ | 0.16 | $740_{\pm45}$ | 0.21 | $\mathbf{738}_{\pm45}$ | 0.21 |
| 5000 | 0.5 | homogenous | 0.5 | all | $804_{\pm40}$ | 0.20 | $783_{\pm42}$ | 0.20 | $\mathbf{680}_{\pm59}$ | 0.26 | $681_{\pm41}$ | 0.27 |
| 5000 | 0.5 | variable | 0.2 | all | $816_{\pm26}$ | 0.18 | $809_{\pm40}$ | 0.16 | $739_{\pm47}$ | 0.21 | $\mathbf{735}_{\pm39}$ | 0.21 |
| 5000 | 0.5 | variable | 0.5 | all | $799_{\pm38}$ | 0.20 | $782_{\pm42}$ | 0.20 | $680_{\pm58}$ | 0.26 | $\mathbf{679}_{\pm46}$ | 0.27 |
| 1000 | 0 | homogenous | 0.2 | agreeing | $787_{\pm24}$ | 0.37 | $754_{\pm40}$ | 0.36 | $\mathbf{747}_{\pm21}$ | 0.34 | $759_{\pm23}$ | 0.31 |
| 1000 | 0 | homogenous | 0.2 | sparse | $826_{\pm41}$ | 0.14 | $822_{\pm55}$ | 0.15 | $\mathbf{782}_{\pm48}$ | 0.17 | $786_{\pm47}$ | 0.18 |
| 1000 | 0 | homogenous | 0.2 | standard | $791_{\pm25}$ | 0.17 | $772_{\pm37}$ | 0.19 | $759_{\pm37}$ | 0.20 | $\mathbf{745}_{\pm23}$ | 0.20 |
| 1000 | 0 | homogenous | 0.2 | unfair | $790_{\pm17}$ | 0.04 | $782_{\pm64}$ | 0.04 | $\mathbf{733}_{\pm67}$ | 0.05 | $742_{\pm56}$ | 0.05 |
| 1000 | 0 | homogenous | 0.5 | agreeing | $795_{\pm11}$ | 0.41 | $724_{\pm52}$ | 0.39 | $\mathbf{704}_{\pm59}$ | 0.40 | $713_{\pm30}$ | 0.37 |
| 1000 | 0 | homogenous | 0.5 | sparse | $815_{\pm17}$ | 0.18 | $796_{\pm104}$ | 0.20 | $\mathbf{728}_{\pm88}$ | 0.22 | $730_{\pm71}$ | 0.24 |
| 1000 | 0 | homogenous | 0.5 | standard | $764_{\pm30}$ | 0.23 | $746_{\pm44}$ | 0.24 | $730_{\pm39}$ | 0.27 | $\mathbf{724}_{\pm47}$ | 0.27 |
| 1000 | 0 | homogenous | 0.5 | unfair | $758_{\pm33}$ | 0.04 | $747_{\pm83}$ | 0.06 | $\mathbf{697}_{\pm77}$ | 0.06 | $699_{\pm63}$ | 0.07 |
| 5000 | 0 | homogenous | 0.2 | agreeing | $807_{\pm51}$ | 0.37 | $754_{\pm31}$ | 0.36 | $\mathbf{748}_{\pm23}$ | 0.34 | $760_{\pm19}$ | 0.31 |
| 5000 | 0 | homogenous | 0.2 | sparse | $824_{\pm39}$ | 0.14 | $819_{\pm56}$ | 0.15 | $\mathbf{782}_{\pm53}$ | 0.17 | $787_{\pm47}$ | 0.18 |
| 5000 | 0 | homogenous | 0.2 | standard | $813_{\pm40}$ | 0.17 | $770_{\pm35}$ | 0.18 | $760_{\pm31}$ | 0.20 | $\mathbf{745}_{\pm33}$ | 0.20 |
| 5000 | 0 | homogenous | 0.2 | unfair | $787_{\pm21}$ | 0.03 | $781_{\pm63}$ | 0.04 | $\mathbf{729}_{\pm69}$ | 0.04 | $740_{\pm53}$ | 0.05 |
| 5000 | 0 | homogenous | 0.5 | agreeing | $795_{\pm18}$ | 0.41 | $721_{\pm55}$ | 0.39 | $\mathbf{698}_{\pm62}$ | 0.41 | $704_{\pm32}$ | 0.37 |
| 5000 | 0 | homogenous | 0.5 | sparse | $805_{\pm33}$ | 0.18 | $799_{\pm99}$ | 0.20 | $\mathbf{731}_{\pm87}$ | 0.22 | $732_{\pm71}$ | 0.24 |
| 5000 | 0 | homogenous | 0.5 | standard | $792_{\pm25}$ | 0.22 | $745_{\pm48}$ | 0.24 | $731_{\pm38}$ | 0.28 | $\mathbf{723}_{\pm52}$ | 0.27 |
| 5000 | 0 | homogenous | 0.5 | unfair | $749_{\pm24}$ | 0.04 | $745_{\pm81}$ | 0.06 | $\mathbf{696}_{\pm72}$ | 0.07 | $699_{\pm62}$ | 0.07 |

Table 2: % of 1 to 1 comparisons throughout capacity constraints settings

| | OvA | DeCCaF$_{greedy}$ | DeCCaF$_{linear}$ | ReL |
|---|---|---|---|---|
| OvA beats | – | 0.05 | 0.04 | 0.65 |
| DeCCaF$_{greedy}$ beats | 0.95 | – | 0.44 | 0.99 |
| DeCCaF$_{linear}$ beats | 0.96 | 0.56 | – | 0.99 |
| ReL beats | 0.35 | 0.01 | 0.01 | - |

of the inner $\Phi$-risk for each $i$th binary classification problem, $g_i^*$. As such, in a scenario where only one expert's prediction is associated with each instance, each binary classifier $g_{\perp,j}$ can be trained independently of the others. By training each binary classifier $g_{\perp,j}$ with the subset of the training sample containing expert $j$'s predictions, we obtain the best possible estimates of each pointwise inner $\Phi$-risk minimizer $g_i^*$ given the available data. To adapt the OvA method to a cost sensitive scenario, we can again use the rescaling approach detailed in section 3.2, minimizing the expected misclassification cost. In our case, we are only interested in the rejector functions, selecting the optimal assignment as $\arg\max_{j\in\{1,...,J\}} g_{\perp,j}(\boldsymbol{x})$.

**Rejection Learning**   For a non L2D baseline, we use the same form of rejection learning that we assume is deployed during months 4 to 7 of our dataset. This approach consists in randomly selecting an expert to make a decision on a model alert.

## 5.2   DEFERRAL SYSTEM TRAINING

In order to train and evaluate our assignment systems, we must first define the cost structure, in order to determine $\lambda$. To derive the value of $\lambda$ from our Neyman-Pearson criterion, we follow the approach detailed in Section A of the Appendix, allowing us to establish a relationship between the target false positive rate and the misclassification cost ratio, which is calculated to be $\lambda = 0.057$. We assume label positive instances have a misclassification cost of 1, while label negative instances have a misclassification cost of $\lambda$.

Having the value of $\lambda$ we can train both our assignment systems. To train the OvA classifiers, as well as the DeCCaF human expertise model, we again use LightGBM, following the process detailed in Sections 5.1 and 3.2. Details on the training process and hyperparameter selection are available in Sections D.1 and D.2 of the appendix. Results regarding the human expertise model and OvA classifiers' performance are shown in Sections D.1.1 and D.2.1, respectively. Further comparisons between these human behavior modelling methods are reported in section D.3, demonstrating the advantages of our approach in cost-sensitive scenarios.

## 5.3   COST-SENSITIVE PERFORMANCE

Having obtained the OvA classifiers and the human expertise model, we now defer under capacity constraints. In Section 3.4 we describe how this is done in DeCCaF. To defer under capacity constraints with the OvA method, we proceed similarly to Verma et al. (2023), by considering the maximum out of the rejection classifiers' predictions. Should the capacity of the selected expert be exhausted, deferral is done to the second highest scoring expert, and so on.

Table 1 displays the test set performance of the assignment systems, as measured by the misclassification cost, calculated as in Equation 5. The results are displayed for several configurations of capacity constraint settings. Each row in the table correspondes to the average of the deferral systems trained on 5 different training sets, tested over a set of capacity constraints generated with the same properties, with 95% confidence intervals. The results demonstrate that the L2D approaches tend to outperform random assignment, showing success in modeling and ranking expert correctness. It can be seen that, on average, both versions of DeCCaF outperform our adaptation of the OvA approach. As expected, DeCCaF$_{\text{linear}}$ frequently outperforms the greedy approach, as it better estimates the optimal set of assignments. The confidence intervals show that the grouping of the cases throughout batches has significant impact on the results of the deferral system, although this is often less evident in DeCCaF$_{\text{linear}}$. Although the confidence intervals intercept, when comparing each individual result, in Table 2 we can see that DeCCaF$_{\text{linear}}$ outperforms ReL and OvA on a vast majority of cases, and that it also outperforms the greedy variant of our method.

## 6   CONCLUSIONS AND FUTURE WORK

In this work we present a novel approach to multi-expert learning to defer, proposing a new model architecture and optimization objective, suitable to cost-sensitive scenarios and compatible with individual human capacity constraints. We have demonstrated that this method outperforms the baselines in a realistic, cost-sensitive fraud detection task.

For future work, we aim to test these systems in scenarios where ML classifier and humans are jointly considered for deferral, instead of having the ML model automatically deferring the instances to be reviewed by human, as is the case in financial fraud prevention. We also plan on testing the addition of fairness incentives to our misclassification cost function, to study the impact this system may have in ensuring fairness. We hope this work promotes further research into L2D methods in realistic settings that consider the key limitations that inhibit the adoption of previous methods in real-world applications.

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

## A  NEYMAN-PEARSON CRITERION AND COST-SENSITIVE LEARNING

In our task, the optimization goal is expressed by a Neyman-Pearson criterion, aiming to maximize recall subject to a fixed FPR of 5%. This criterion is chosen due to the difference between error types. In bank account fraud prevention, the consequence of committing a false positive mistake, that is, rejecting a legitimate application, must be weighed against the cost of a false negative mistake, that is, accepting a fraudulent application. To apply our approach, we must convert this criterion into a cost structure. In a binary classification task, the possible outcomes of a prediction are the elements of a confusion matrix; true negative, false negative, false positive, true positive: [TN, FN, FP, TP]. Over a set of assignments, the loss is given by the total number of each outcome multiplied by its cost $\{c_{00}, c_{01}, c_{10}, c_{11}\}$, respectively:

$$
\begin{aligned}
L &= c_{00}\text{TN} + c_{01}\text{FN} + c_{10}\text{FP} + c_{11}\text{TP} \\
&= c_{00}(\text{LP} - \text{FN}) + c_{01}\text{FN} + c_{10}\text{FP} + c_{11}(\text{LN} - \text{FP}) \\
&= (c_{01} - c_{00})\text{FN} + (c_{10} - c_{11})\text{FP} + c_{00}\text{LP} + c_{11}\text{LN}
\end{aligned}
$$

where LP denotes the total number of label positives and LN denotes the total number of label negatives. Since LP and LN are constants, these do not affect resulting rankings when comparing methods. As such, $c_{00}$ and $c_{11}$ can be set to zero. Doing so, we reach an equation of the form:

$$
L = \lambda\text{FP} + \text{FN} \quad \text{with} \quad \lambda = \frac{c_{10}}{c_{01}} \tag{4}
$$

The predicted loss associated with deferring a case $\mathbf{X}_i$ to an expert $e$ corresponds to the sum of the predicted probability of outcomes weighted by their cost, arriving at the loss function used in our methods.

$$
L(\mathbf{X}_i, e) = \lambda\hat{\mathbb{P}}(\text{FP}) + \hat{\mathbb{P}}(\text{FN}) \tag{5}
$$

However, in our task, we do not have access to the values of $c_{10}$ and $c_{01}$. We must establish a relationship between the value of $\lambda$ and the desired Neyman-Pearson criterion. According to Elkan (2001), we can establish a relationship between the ideal threshold of a binary classifier and the misclassification costs. For a given instance, the ideal classification is the one that minimizes the expected loss. As such, the optimal prediction for any given instance $\mathbf{x_i}$ is 1 only if the expected cost of predicting 1 is less than, or equal to the expected cost of predicting 0, which is equivalent to:

$$
(1 - p)c_{10} \leq pc_{01} \tag{6}
$$

Where $p = P(y = 1|\mathbf{x}_i)$, that is, the probability that $x_i$ belongs to the positive class. An estimation of the value of $p$ is given by our classifier, in the form of the model score output for a given instance, which is an estimate of the probability that $x_i$ belongs to class 1. In the case where the inequality is in fact an equality, then predicting either class is optimal. As such, the decision threshold $t$ for making optimal decisions leads us to a value for $\lambda$:

$$
(1 - t)c_{10} = tc_{01} \Leftrightarrow \lambda = \frac{t}{1 - t} \tag{7}
$$

As the optimal threshold $t$ for our ML model was chosen such that the Neyman-Pearson criterion is met, we now may plug the value of $t$ into this equation, obtaining the theoretical value of lambda for our optimization goal. We chose to use this theoretically defined value such that the same value of $\lambda$ is used across all methods' loss functions.

However, as shown by Sheng & Ling (2006), choosing the theoretical threshold does not always work in practice. The authors show that testing several values for $\lambda$ in validation works best. Secondly, the value of $\lambda$ obtained through this method depends on the classifier trained. A different classifier would yield another value for the optimal threshold according to the Neyman-Pearson criterion, which would lead to a different $\lambda$, despite the task being the same.

## B  SYNTHETIC EXPERT GENERATION METHOD

### B.1  EXPERT DESIDERATA

**Feature and AI assistant dependence**  When a decision is made by an expert, it is assumed that they will base themselves on information related to the instance in question. Therefore, we expect experts to be dependent on an instance's features in order to make an informed decision.

However, in some real world deferral systems (Amarasinghe et al., 2022; De-Arteaga et al., 2020), the instance's features are accompanied by an AI model's score, representing the model's estimate of the probability that said instance belongs to the positive class. The aim of presenting the model score to an expert is to provide them with extra information, as well as possibly expediting the decision process. It has been shown that, in this scenario, expert's performance can be impacted by presenting the model's score when deferring a case to an expert (Amarasinghe et al., 2022; De-Arteaga et al., 2020; Levy et al., 2021). Note that, in this setting, a classifier trained for our task exists independently of the assignment system implemented. This approach is applicable to deferral systems such as the one proposed by Raghu et al. (2019a), who argue for the training of a separate human behaviour prediction model (Raghu et al., 2019a;b). In the L2D framework, however, the main classifier is trained jointly with the deferral system. As such, this framework allows users to generate experts with or without dependence on a separate ML classifier's score.

**AI assistance and algorithmic bias** Should the generated experts use an AI assistant, we expect experts not to be in perfect agreement with the model, due to the assumption that humans and models have complementary strengths and weaknesses (De-Arteaga et al., 2020; Dellermann et al., 2019). As such, we would assume humans and AI perform better than one another in separate regions of the feature space, enabling an assignment system to obtain better performance than either the expert team or the model on their own. The degree of "model dependence", or "algorithmic bias" (Alon-Barkat & Busuioc, 2023),varies between humans, as measured by the model's impact on a human's performance (Jacobs et al., 2021; Inkpen et al., 2022). As such, our synthetic expert team may also exhibit varying levels of dependence on the model score.

**Varied Expert Performance** In order for our team of experts to be realistic, it is important that these exhibit varying levels of overall performance. Experts within a field have been shown to have varying degrees of expertise, with some being outperformed by ML models (Goel et al., 2021; Gulshan et al., 2016). As such human decision processes can be expected to be varied even amongst a team of experts.

**Predictability and Consistency** It is a common assumption that, when making a decision, experts follow an internal process based on the available information. However, it is also known that highly educated and trained individuals are still subject to flaws that are inherent to human decision making processes, one of these being inconsistency. When presented with similar cases, at different times, experts may perform drastically different decisions (Danziger et al., 2011; Grimstad & Jørgensen, 2007). Therefore we can expect a human's decision making process not to be entirely deterministic.

**Human Bias and Unfairness** It is also important to consider the role that the assignment system can play in mitigating unfairness. If an expert can be determined to be particularly unfair with respect to a given protected attribute, the assignment system can learn not to defer certain cases to that expert. In order to test the fairness of the system as a whole, it is may be useful to create a team comprised of individuals with varying propensity for unfair decisions.

## B.2 EXPERT PARAMETER SAMPLING

As stated in section 4.1, we define the expert's probabilities of error, for any given instance, as a function of its features, $\boldsymbol{x}_i$, and an ML model score $m(\boldsymbol{x}_i)$:

$$
\begin{cases}
\mathbb{P}(\hat{y}_i = 1|y_i = 0, \boldsymbol{x}_i, M) = \sigma\left(\beta_0 - \alpha \frac{\boldsymbol{w}\cdot\boldsymbol{x}_i + w_M M(\boldsymbol{x}_i)}{\sqrt{\boldsymbol{w}\cdot\boldsymbol{w} + w_M^2}}\right) \\
\mathbb{P}(\hat{y}_i = 0|y_i = 1, \boldsymbol{x}_i, M) = \sigma\left(\beta_1 + \alpha \frac{\boldsymbol{w}\cdot\boldsymbol{x}_i + w_M M(\mathbf{x}_i)}{\sqrt{\boldsymbol{w}\cdot\boldsymbol{w} + w_M^2}}\right)
\end{cases}, \quad M(\mathbf{x}_i) = \begin{cases} \frac{m(\boldsymbol{x}_i) - t}{2t}, & m \leq t \\ \frac{m(\boldsymbol{x}_i) - t}{2(1-t)}, & m > t \end{cases}.
$$

$$(8)$$

Where $\sigma$ denotes a sigmoid function and $M$ is a transformed version of the original model score $m \in [0, 1]$. Each expert's probabilities of the two types of error are parameterized by five parameters: $\beta_0, \beta_1, \alpha, \boldsymbol{w}$ and $w_M$. The sampling process of each parameter is done as follows:

**Feature Dependence Weights Generation** To define $\boldsymbol{w}$ for a given expert, we sample each component from a "Spike and Slab" prior (Mitchell & Beauchamp, 1988). A spike and slab prior is a generative model in which a random variable $u$ either attains some fixed value $v$, called the *spike*,

or is drawn from another prior $p_{\text{slab}}$, called the *slab*. In our case, we set $v = 0$. That is, $u$ is either zero, or drawn from the slab density $N(\mu_w, \sigma_w)$, where $\mu_w, \sigma_w$ are defined by the user. To sample the values of $w_i$, we first sample a Bernoulli latent variable $Z \sim Ber(\theta)$ to select if $w_i$ is sampled from the *spike* or the *slab*. If $Z = 0$, $w_i$ attains the fixed value $v = 0$, if $Z = 1$, $w_i$ is drawn from the slab density $p_{\text{slab}}$. As such, the spike and slab prior induces sparsity unless $\theta = 1$, allowing for the generation of experts whose probabilities of error are swayed by a varying number of features, by changing the value of $\theta$. The distribution of $w_M \sim N(\mu_M, \sigma_M)$ can be defined separately to allow for control of the expert's model-dependency. Should there be a protected attribute, the user is also able to define $\mu_p, \sigma_p$, in order to impose fairness/unfairness on the simulated experts' predictions.

**Controlling Variability and Expert's consistency**   While the weight vector controls the relative influence that each feature has on the probability of error, parameter $\alpha$, in turn, controls the global magnitude of this influence. For $\alpha = 0$, the probability of error would be identical for all instances. In turn, for very large $\alpha$, the probability would saturate at the extremes of the codomain of the sigmoid function, 0 or 1, resulting in a deterministic decision-making process. As such, $\alpha \sim N(\mu_\alpha, \sigma_\alpha)$ can be chosen such that a wide variety of probability of errors exist throughout the feature space.

**Controlling Expert Performance**   In a binary classification task, two metrics often used to evaluate a classifier's performance are the false positive rate (FPR) and false negative rate (FNR). Our framework allows users to set the distributions from which the expert's target performance metrics are sampled, $T_{\text{FPR}} \sim N(\mu_{\text{FPR}}, \sigma_{\text{FPR}})$ and $T_{\text{FNR}} \sim N(\mu_{\text{FNR}}, \sigma_{\text{FNR}})$. Our framework then calculates the value of $\beta_0$ and $\beta_1$ to obtain the desired performance. This allows a user to avoid adjusting $\beta_0$ and $\beta_1$ iteratively. For details on how $\beta_0$ and $\beta_1$ are calculated, see section B.3 of the Appendix.

**Feature Preprocessing**   For our simulation of experts, the feature space is transformed as follows. Numeric features in $\mathbf{X}$ are transformed to quantile values, and shifted by $-0.5$, resulting in features with values in the $[-0.5, 0.5]$ interval. This ensures that the features impact the probability of error independently of their original scale. Categorical features are target-encoded, that is, encoded into non-negative integers by ascending order of target prevalence. These values are divided by the number of categories, so that they belong to the $[0, 1]$ interval, and shifted so that they have zero mean.

These distributions are defined in a group-wise fashion in order to allow for the generation of several different expert groups.

### B.3   CALCULATION OF $\beta_0$ AND $\beta_1$

In our framework, the parameter $\beta_0$ and $\beta_1$ serve the purpose of controlling each expert's FPR and FNR, separately. However, we wish to allow the expert to define the target performance metrics $T_{\text{FPR}}$ and $T_{\text{FNR}}$. In other words, we want the expected value of the FPR when generating an expert to be equal to $T_{\text{FPR}}$. If the value of the weights $\mathbf{w}$ and $w_M$, as well as $\alpha$ are sampled prior to defining $\beta_0$ and $\beta_1$, an expert's empirical false positive rate, $\text{FPR}_e$ depends only on $\beta_1$:

$$\text{FPR}_e(\beta_1) = \frac{1}{N} \sum_i \sigma\left(\beta_1 + \alpha \frac{\boldsymbol{w}.\boldsymbol{x}_i}{||\boldsymbol{w}||}\right) . \tag{9}$$

Note that, for notational simplicity we set $w_M = 0$ but the result is similar when $w_M \neq 0$. It is then possible to show that the function $\text{FPR}_e(\beta_1)$ is monotonically increasing,

$$\frac{\partial \text{FPR}_e}{\partial \beta_1} = \frac{1}{N} \sum_i \sigma\left(\beta_1 + \alpha \frac{\boldsymbol{w}.\boldsymbol{x}_i}{||\boldsymbol{w}||}\right)\left(1 - \sigma\left(\beta_1 + \alpha \frac{\boldsymbol{w}.\boldsymbol{x}_i}{||\boldsymbol{w}||}\right)\right) > 0 \quad \text{for} \quad \beta \in \mathbb{R} . \tag{10}$$

Since the function is monotonically increasing,and in a bounded to the interval $]0, 1[$, then, for any target false positive rate $T_{\text{FPR}}$, then the following equation has an unique solution:

$$\text{FPR}_e(\beta_1) - T_{\text{FPR}} = 0 \quad \text{for} \quad T_{\text{FPR}} \in ]0, 1[ . \tag{11}$$

A similar reasoning applies for $\beta_0$ and $T_{\text{FNR}}$. Finally, we can control an expert's FPR and FNR by solving these equations for $\beta_1$ and $\beta_0$. To solve these equations, a partition of the dataset is utilized to calculate the empirical value for each rate. Due to the monotonous nature of the function, and the uniqueness of the solution, we solve it through a bisection method (Burden & Faires, 1985).

## C    TESTBED

### C.1    ML MODEL

As detailed in Section 4.1, our ML Model is a LightGBM (Ke et al., 2017) classifier. The model was trained on the first 3 months of the BAF dataset, and validated on the fourth month. The model is trained by minimizing binary cross-entropy loss. The choice of hyperparameters is defined through Bayesian search (Akiba et al., 2019) on an extensive grid, for 100 trials, with validation done on the 4th month, where the optimization objective is to maximize recall at 5% false positive rate in validation. In Table 3 we present the hyperparameter search space used, as well as the parameters of the selected model.

Table 3: ML Model: LightGBM hyperparameter search space

| Hyperparameter | Values or Interval | Distribution | Selected value |
|---|---|---|---|
| boosting_type | "goss" | | "goss" |
| enable_bundle | False | | False |
| n_estimators | [50,5000] | Logarithmic | 94 |
| max_depth | [2,20] | Uniform | 2 |
| num_leaves | [10,1000] | Logarithmic | 145 |
| min_child_samples | [5,500] | Logarithmic | 59 |
| learning_rate | [0.01, 0.5] | Logarithmic | 0.3031421 |
| reg_alpha | [0.0001, 0.1] | Logarithmic | 0.0012637 |
| reg_lambda | [0.0001, 0.1] | Logarithmic | 0.0017007 |

This model yielded a recall of 57.9% in validation, for a threshold $t = 0.050969$, defined to obtain a 5% FPR in validation. In the deployment split (months 4 to 8), used to train and test our assignment system, the model yields a recall of $M_{\text{TPR}} = 52.1\%$, using the same threshold. The false negative rate of the model, later used to generate our synthetic experts, is then $M_{\text{FNR}} = 47.9\%$. In Figure 3 we present the ROC curve for our model, calculated in the deployment split.

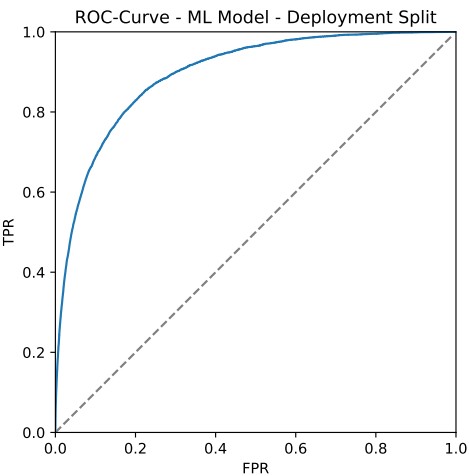

Figure 3: ROC-Curve - ML Model - Deployment Split

### C.2    SYNTHETIC EXPERT TEAM SETTINGS

We created a team comprised of "standard","model-agreeing", "unfair" and "sparse" experts. In Table 4 we present the distribution values defined for each expert group. Sampling from these

Table 4: Expert Group Properties

| Parameter | Group | | | |
|---|---|---|---|---|
| | Standard | Model Agreeing | Unfair | Sparse |
| n | 20 | 10 | 10 | 10 |
| group_seed | 1 | 2 | 3 | 1 |
| $\mu_w$ | 0 | 0 | 0 | 0 |
| $\sigma_w$ | 1 | 1 | 1 | 1 |
| $\theta$ | 0.3 | 0.3 | 0.3 | 0.1 |
| $\mu_M$ | -2 | -6 | -2 | -2 |
| $\sigma_M$ | 0.5 | 0.5 | 0.5 | 0.5 |
| $\mu_p$ | -1 | 0 | -4 | -1 |
| $\sigma_p$ | 0.1 | 0.1 | 0.3 | 0.1 |
| $\mu_\alpha$ | 4 | 12 | 4 | 4 |
| $\sigma_\alpha$ | 0.2 | 0.5 | 0.1 | 0.2 |
| $\mu_{\text{FNR}}$ | $M_{\text{FNR}} - \sigma_{\text{FNR}}$ | $M_{\text{FNR}} - \sigma_{\text{FNR}}$ | $M_{\text{FNR}} - \sigma_{\text{FNR}}$ | $M_{\text{FNR}} - \sigma_{\text{FNR}}$ |
| $\sigma_{\text{FNR}}$ | 0.05 | 0.05 | 0.05 | 0.05 |
| $\mu_{\text{FPR}}$ | $M_{\text{FPR}} - \sigma_{\text{FPR}}$ | $M_{\text{FPR}} - \sigma_{\text{FPR}}$ | $M_{\text{FPR}} - \sigma_{\text{FPR}}$ | $M_{\text{FPR}} - \sigma_{\text{FPR}}$ |
| $\sigma_{\text{FPR}}$ | 0.01 | 0.01 | 0.01 | 0.01 |

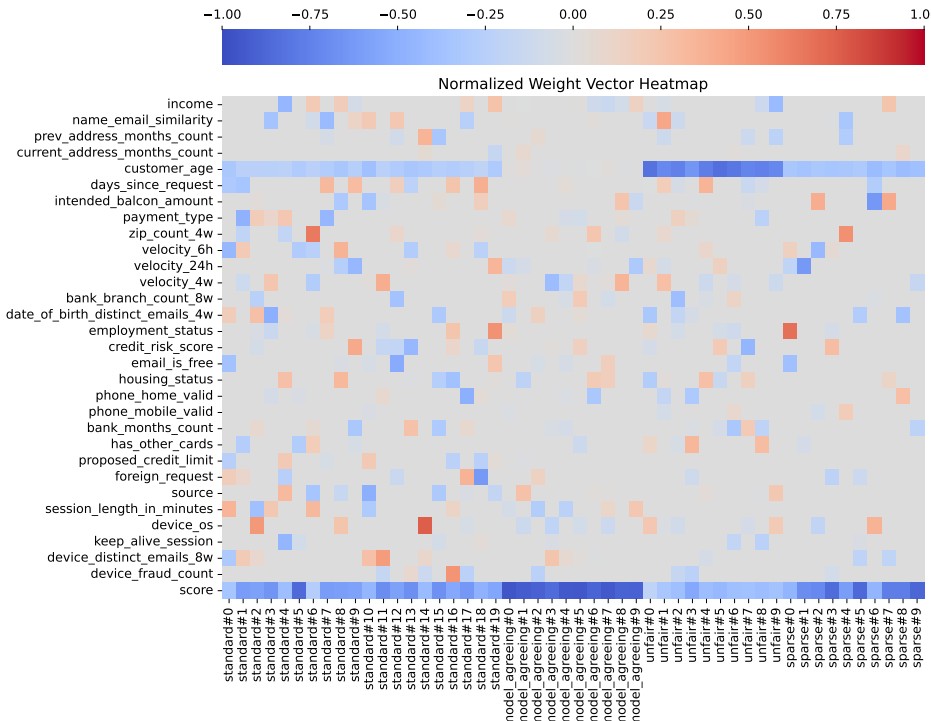

Figure 4: Normalized Feature weights

distribution values, each expert's parameters - $T_{\text{FPR}}, T_{\text{FNR}}, \alpha, w_M$ and $w_p$ - are presented in Table 5. Using these properties, the feature weight vector $\mathbf{w}$ is also sampled for each expert.

To display each synthetic expert's feature dependence, the values of the normalized feature weights comprising the vector $\boldsymbol{w}$ are displayed in in Figure 4, in a heatmap. As intended, the dominant feature for the unfair experts is the customer's age, which is our protected attribute, while the model agreeing experts are mostly influenced by the model score. Our standard experts display a balanced weight distribution across features, while the sparse experts show dependency with less features.

Table 5: Individual Expert's Parameters

| Expert | $T_{\text{FPR}}$ | $T_{\text{FNR}}$ | $\alpha$ | $w_p$ | $w_M$ |
|---|---|---|---|---|---|
| | | | Parameters | | |
| standard#0 | 0.392 | 0.038 | 3.96 | -1.11 | -1.19 |
| standard#1 | 0.492 | 0.038 | 3.82 | -0.89 | -2.31 |
| standard#2 | 0.455 | 0.042 | 3.85 | -0.91 | -2.26 |
| standard#3 | 0.414 | 0.044 | 4.34 | -0.95 | -2.54 |
| standard#4 | 0.454 | 0.042 | 4.01 | -0.91 | -1.57 |
| standard#5 | 0.426 | 0.041 | 3.87 | -1.07 | -3.15 |
| standard#6 | 0.486 | 0.033 | 4.04 | -1.01 | -1.13 |
| standard#7 | 0.505 | 0.044 | 4.42 | -1.09 | -2.38 |
| standard#8 | 0.539 | 0.041 | 4.02 | -1.03 | -1.84 |
| standard#9 | 0.359 | 0.051 | 4.12 | -0.95 | -2.12 |
| standard#10 | 0.357 | 0.052 | 4.06 | -1.07 | -1.27 |
| standard#11 | 0.404 | 0.042 | 3.93 | -1.04 | -3.03 |
| standard#12 | 0.437 | 0.036 | 3.77 | -1.07 | -2.16 |
| standard#13 | 0.473 | 0.034 | 3.93 | -1.08 | -2.19 |
| standard#14 | 0.445 | 0.044 | 3.96 | -1.07 | -1.43 |
| standard#15 | 0.328 | 0.041 | 4.12 | -1.00 | -2.55 |
| standard#16 | 0.414 | 0.037 | 4.17 | -1.11 | -2.09 |
| standard#17 | 0.471 | 0.040 | 4.19 | -0.98 | -2.44 |
| standard#18 | 0.441 | 0.034 | 4.06 | -0.83 | -1.98 |
| standard#19 | 0.467 | 0.047 | 4.18 | -0.93 | -1.71 |
| model_agreeing#0 | 0.416 | 0.037 | 11.56 | 0.06 | -6.21 |
| model_agreeing#1 | 0.541 | 0.048 | 11.92 | 0.23 | -6.03 |
| model_agreeing#2 | 0.308 | 0.021 | 12.13 | 0.00 | -7.07 |
| model_agreeing#3 | 0.435 | 0.057 | 11.51 | -0.11 | -5.18 |
| model_agreeing#4 | 0.448 | 0.055 | 11.83 | 0.05 | -6.90 |
| model_agreeing#5 | 0.497 | 0.037 | 11.88 | -0.06 | -6.42 |
| model_agreeing#6 | 0.454 | 0.046 | 11.68 | 0.00 | -5.75 |
| model_agreeing#7 | 0.387 | 0.040 | 11.41 | 0.12 | -6.62 |
| model_agreeing#8 | 0.429 | 0.032 | 11.29 | -0.07 | -6.53 |
| model_agreeing#9 | 0.456 | 0.041 | 11.92 | 0.00 | -6.45 |
| unfair#0 | 0.467 | 0.024 | 3.88 | -4.39 | -1.11 |
| unfair#1 | 0.528 | 0.046 | 3.98 | -3.73 | -1.78 |
| unfair#2 | 0.367 | 0.036 | 4.15 | -3.74 | -1.95 |
| unfair#3 | 0.398 | 0.023 | 4.02 | -3.49 | -2.93 |
| unfair#4 | 0.389 | 0.034 | 3.90 | -3.98 | -2.14 |
| unfair#5 | 0.308 | 0.034 | 3.93 | -4.12 | -2.18 |
| unfair#6 | 0.383 | 0.031 | 4.06 | -4.16 | -2.04 |
| unfair#7 | 0.378 | 0.040 | 3.98 | -4.46 | -2.31 |
| unfair#8 | 0.486 | 0.018 | 3.92 | -3.71 | -2.02 |
| unfair#9 | 0.423 | 0.037 | 3.98 | -4.33 | -2.24 |
| sparse#0 | 0.395 | 0.038 | 3.78 | -0.85 | -1.19 |
| sparse#1 | 0.409 | 0.031 | 4.23 | -1.21 | -2.31 |
| sparse#2 | 0.395 | 0.033 | 4.18 | -1.03 | -2.26 |
| sparse#3 | 0.387 | 0.057 | 4.10 | -1.04 | -2.54 |
| sparse#4 | 0.396 | 0.041 | 4.18 | -0.89 | -1.57 |
| sparse#5 | 0.429 | 0.034 | 3.86 | -1.11 | -3.15 |
| sparse#6 | 0.373 | 0.042 | 3.98 | -1.02 | -1.13 |
| sparse#7 | 0.441 | 0.061 | 3.81 | -1.09 | -2.38 |
| sparse#8 | 0.512 | 0.041 | 3.95 | -1.00 | -1.84 |
| sparse#9 | 0.466 | 0.046 | 4.11 | -0.94 | -2.12 |

### C.2.1 GENERATED PROBABILITIES

The selected values for $\alpha$ were chosen so that every expert displays a wide range of probabilities of error. In Figures 5 and 6, we display the distribution of probabilities of error throughout instances, for our 3 different expert groups.

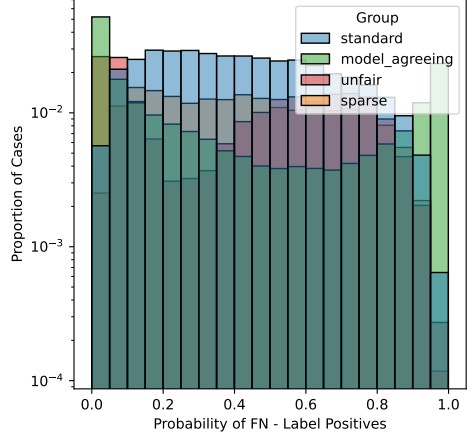

Figure 5: P(FN) - Label Positives

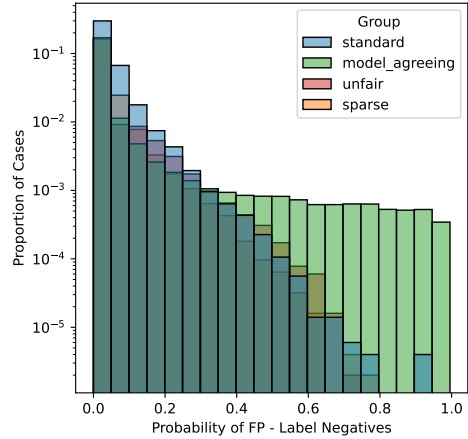

Figure 6: P(FP) - Label Negatives

Each group's distributions are significantly different. The use of a larger value of $\alpha$ for model agreeing experts result in predictions saturating at the leftmost and rightmost bin, nearer the edges of the $[0, 1]$ interval.

### C.2.2 PERFORMANCE

When generating our team of synthetic experts, we aimed to obtain a team where most, but not all experts are better than the ML model. In a cost-sensitive task such as this, defining what "better" means is not trivial, as there is an implicit trade-off between FPR and FNR, imposed by our Neyman-Pearson criterion. In choosing our reported values for $T_{\text{FPR}}, T_{\text{FNR}}$, we aimed to obtain a team where most experts have both a lower FPR and a higher TPR than the model. A plot of each expert's Target FPR and recall when compared to the model is presented in Figure 7. Due to distribution shifts present across months in the bank-account-fraud dataset, the FPR and TPR obtained in the deployment split will deviate from the target values. The values of the FPR and TPR of each expert in comparison to the model, calculated in the deployment split, are presented in Figure 8. To compare the performance of the experts and the ML model using a single metric, we may use the loss function defined in section A, over the entire deployment split.

Table 6: Complementary Expertise: Fraction of instances within each class where expert and model disagree. $y_x$ denotes the label, $m_x$ the model's prediction, and $e_x$ the expert's prediction. The subscript denotes the label/prediction value.

| Expert Group | $y_0, m_0, e_1$ | $y_0, m_0, e_1$ | $y_0, m_0, e_1$ | $y_0, m_0, e_1$ |
|---|---|---|---|---|
| Model Agreeing | 0.155 | 0.067 | 0.027 | 0.022 |
| Sparse | 0.204 | 0.162 | 0.032 | 0.039 |
| Standard | 0.224 | 0.174 | 0.036 | 0.040 |
| Unfair | 0.232 | 0.169 | 0.028 | 0.041 |

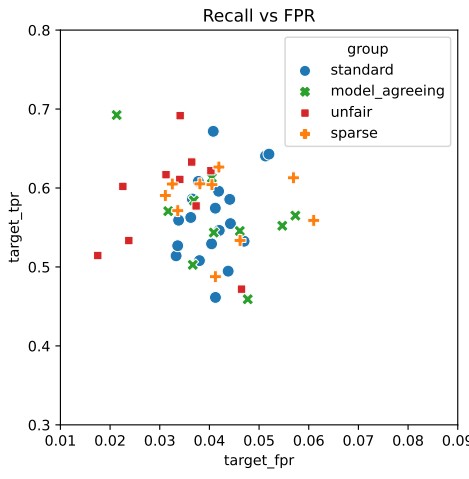

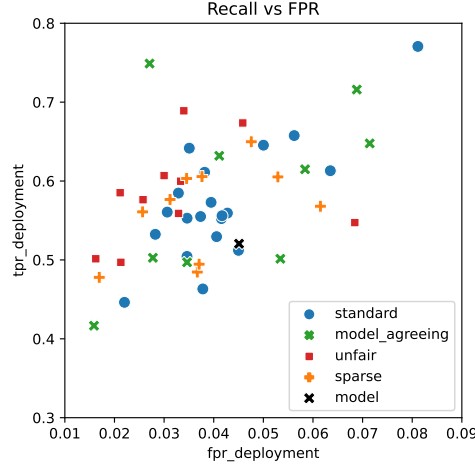

Figure 7: Target TPR and FPR

Figure 8: Deployment TPR and FPR

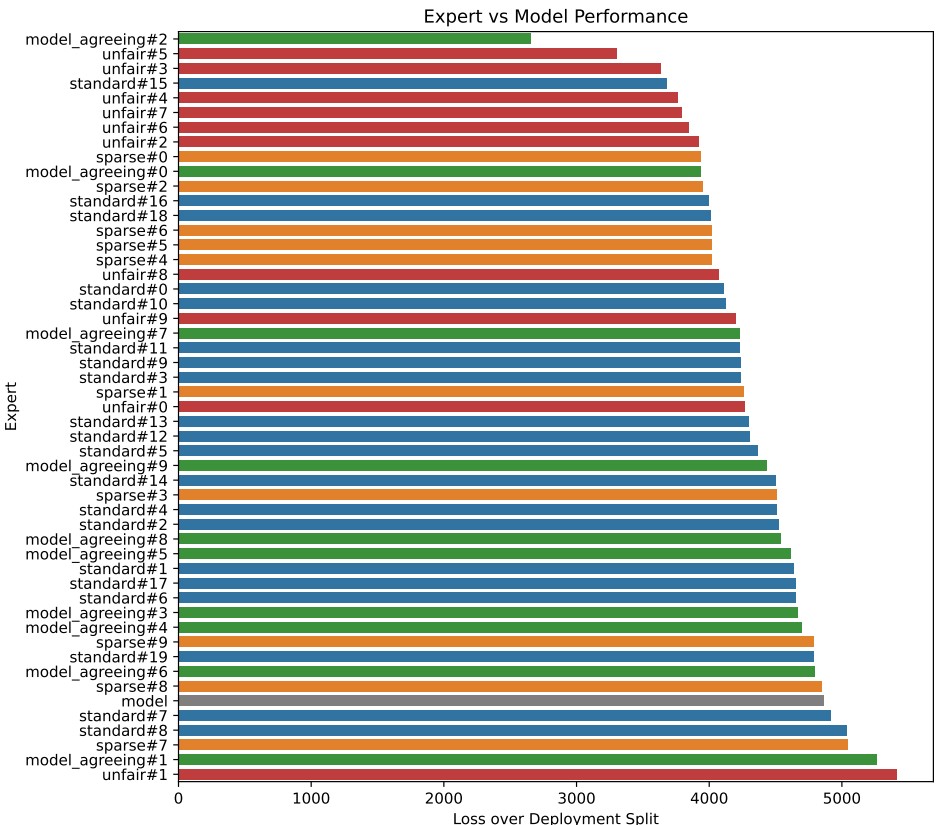

Figure 9: Performance - Misclassification cost across deployment split

### C.2.3 Unfairness and Model-Agreement

To introduce humans with a higher bias against older customers (age $\geq 50$), we created a group of unfair experts. When setting a lower $\mu_p$ for the unfair experts, we expect to increase the probability of committing a FP mistake for older customers, as the customer's age will now be a primary factor in the probability of committing a mistake. The FPR disparities of each expert on the deployment

split are displayed in Figure 10, confirming that our unfair experts exhibit a higher bias against older customers.

Despite the higher average performance of our experts, we expect the model and the experts to have complementary expertise. We test if there is a significant number of instances where the model predicts correctly, and the expert incorrectly, or vice-versa. To do so, we calculate the fraction of instances within each label $y$ where the expert $e$ is correct and the model $M$ is incorrect, or vice versa. These results are presented in Table 6. These show that both the expert and the model have a large amount of cases where disagreement takes place. We can also see that Model Agreeing experts exhibit less disagreement with the model. Using Cohen's Kappa, which aims to measure inter-labeler agreement between two sets of predictions, accounting for differences in prevalence, we can confirm that the model-agreeing experts' predictions are more similar to the model's. In Figure 10 we present the Cohen's Kappa for each expert in relation to the model's predictions.

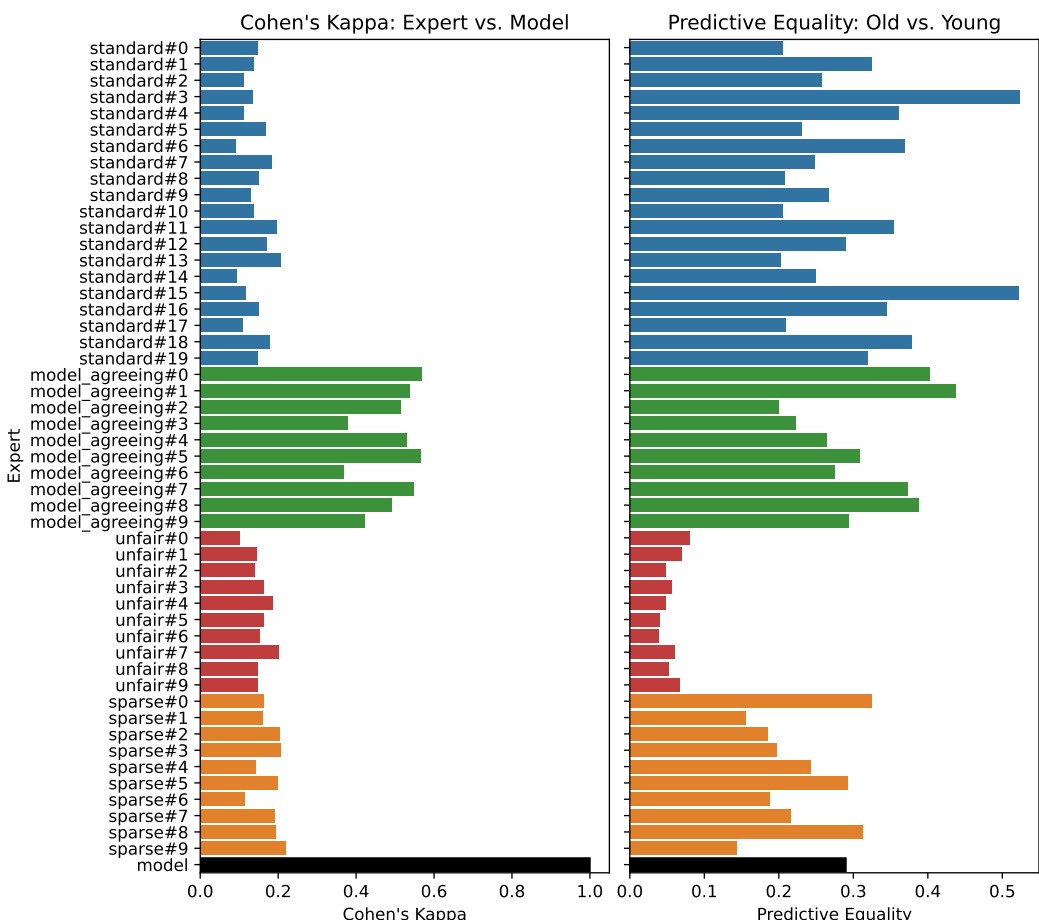

Figure 10: Cohen's Kappa and FPR Disparity across experts

# D HUMAN EXPERTISE MODEL AND OVA CLASSIFIERS

## D.1 TRAINING HUMAN EXPERTISE MODEL

As referred in section 5.2, we assume that misclassifying a label positive instance, that is, an instance belonging to classes 'TP, FN' will have a cost of 1, while misclassifying a label negative instance (classes 'FP', 'TN') will incur a cost of $\lambda$. It is shown by Zadrozny et al. (2003) that rescaling the instances in this way is equivalent to training on data sampled from a different distribution, as stated in equation 1 of the main paper. However, we wish for our probability estimates to be suited

for the original training set distribution, which means the probability estimates of our model must be calibrated. To calibrate the false positive and false negative probability estimates, we reduce the multi-class problem to two binary "One versus Rest" problems, concerning the classification of false positives and false negatives. The probability estimates for each class are then calibrated using Isotonic Regression (Zadrozny & Elkan, 2002).

The choice of hyperparameters is defined through Bayesian search (Akiba et al., 2019) on an extensive grid, for 120 trials (100 startup trials) with validation done on the 7th month, where the optimization objective is to minimize the weighted cross-entropy loss. This optimization objective is chosen to obtain the best possible probability estimates. In Table 3 we present the hyperparameter search space used, as well as the parameters of the selected model. We can see that on four of the five training seeds, the same model hyperparameters were chosen.

Table 7: Expertise Model: LightGBM hyperparameter search space

| Hyperparameter | Values or Interval | Dist. | Selected values | | | | |
| --- | --- | --- | --- | --- | --- | --- | --- |
| | | | Seed 1 | Seed 2 | Seed 3 | Seed 4 | Seed 5 |
| boosting_type | {"dart"} | | "dart" | "dart" | "dart" | "dart" | "dart" |
| enable_bundle | {False,True } | | False | False | False | False | False |
| learning_rate | [0.005, 0.5] | Log | 0.014 | 0.014 | 0.028 | 0.014 | 0.014 |
| min_child_samples | [5,200] | Log | 31 | 31 | 8 | 31 | 31 |
| n_estimators | [200,1000] | Log | 595 | 595 | 286 | 595 | 595 |
| num_leaves | [100,1000] | Log | 103 | 103 | 110 | 103 | 103 |
| reg_alpha | [0.0001, 0.1] | Log | 0.0086 | 0.0086 | 0.0333 | 0.0086 | 0.0086 |
| reg_lambda | [0.0001, 0.1] | Log | 0.0003 | 0.0003 | 0.0989 | 0.0003 | 0.0003 |

### D.1.1 EVALUATION OF HEM IN THE TEST SET

We want to evaluate our model as a predictor of each expert's probability of error. To do so, we take each instance of the test set, and set the assignment as a given expert, obtaining the probability of error of said expert, for each instance in the test set. We can then evaluate our model's performance in an "expert-wise" fashion. As the Human Expertise Model has a multiclass output, we will evaluate the area under the ROC curve for the binary tasks of false negative and false positive identification.

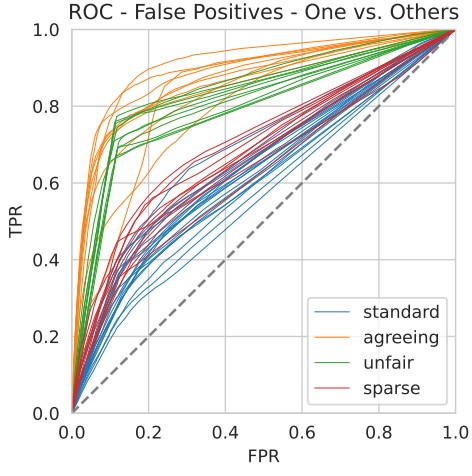
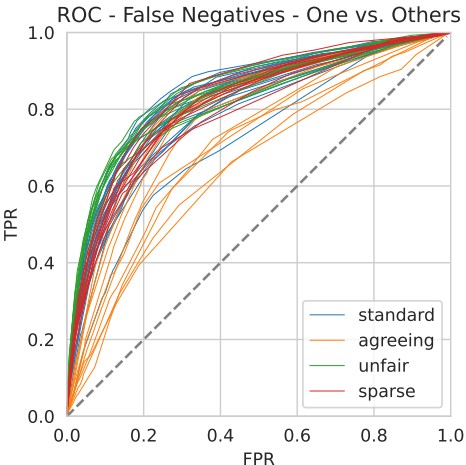

Figure 11: ROC-Curve for FP - Seed 1 - Test    Figure 12: ROC-Curve for FN - Seed 1 - Test

We can see that the OvO ROC curves for the False Positive identification behave as we would expect. The standard experts are the most difficult to predict, as their decision process is most

complex, followed by the sparse experts, who are dependent on less features. The unfair and model agreeing experts are much simpler, being mostly swayed by a single feature, which may explain their higher AUC. We can see that the OvO ROC curves for the False Negative identification are much more similar across experts. Counter-intuitively, the model agreeing experts exhibit the lowest ROC AUC, seeming to be more "unpredictable" when commiting false negative mistakes. The AUC for FN prediction is higher across experts when compared to the FP curves. This is most likely due to the fact that the distribution of probabilities of error is significantly different for label positives and label negatives, as seen in Figures [ref]. We can, however, conclude that our model is successful in capturing the team's behavior.

## D.2 Training OvA Baseline

As mentioned in section 5.1, to train the OvA binary rejection classifiers, we apply the same reweighting technique. To train these classifiers we again utilize the LightGBM algorithm. The hyperparameters were selected similarly to the Human Expertise Model, through Bayesian search on a grid for 120 trials, of which the first 100 are startup trials. The optimization objective for each of these classifiers is to minimize the binary log loss, similarly to Verma et al. (2023). In Table 8 we present the hyperparameter search space. Since 50 models were trained, we do not display the individual values chosen for each.

Table 8: OvA classifiers: LightGBM hyperparameter search space

| Hyperparameter | Values or Interval | Dist. |
|---|---|---|
| boosting_type | {"dart"} | |
| enable_bundle | {False,True } | |
| max_depth | [2,20] | Uni |
| learning_rate | [0.005, 0.5] | Log |
| min_child_samples | [5,200] | Log |
| n_estimators | [200,1000] | Log |
| num_leaves | [100,1000] | Log |
| reg_alpha | [0.0001, 0.1] | Log |
| reg_lambda | [0.0001, 0.1] | Log |

### D.2.1 Evaluating OvA Baseline

The key differences between these classifiers and our Human Expertise Model are the existence of an individual model for each expert, and the fact that we only predict whether the expert is correct or not, without differentiating false positives from false negatives, or true positives from true negatives. We can then evaluate the ROC AUC of each of these classifiers:

## D.3 OvA Rejection Classifiers vs Human Expertise Model

To estimate expert error utilizing our HEM, we add the values of the probability that an expert commits a false positive and a false negative. In this fashion we can compare the estimates of expert correctness of our proposed method and the OvA approach. To do so, we display the value of the ROC AUC for each of the expert's rejection classifiers, as wel as the value of the ROC AUC for each of the expert's when predicting their probability of error with the HEM.

Table 9: AUC Comparison OvA versus human expertise model - averaged across seeds

| | Human Expertise Model | | | OvA |
|---|---|---|---|---|
| Expert | $AUC_{FN}$ | $AUC_{FP}$ | $AUC_{Expert\ Error}$ | $AUC_{Expert\ Error}$ |
| standard#0 | 0.833 | 0.648 | **0.682** | 0.642 |
| standard#1 | 0.838 | 0.606 | **0.646** | 0.576 |
| standard#2 | 0.844 | 0.609 | **0.633** | 0.620 |
| standard#3 | 0.799 | 0.587 | **0.602** | 0.575 |
| standard#4 | 0.846 | 0.607 | **0.633** | 0.595 |
| standard#5 | 0.815 | 0.656 | **0.677** | 0.623 |
| standard#6 | 0.857 | 0.572 | **0.631** | 0.581 |
| standard#7 | 0.824 | 0.663 | **0.680** | 0.639 |
| standard#8 | 0.835 | 0.659 | **0.687** | 0.638 |
| standard#9 | 0.847 | 0.621 | **0.674** | 0.611 |
| standard#10 | 0.819 | 0.660 | **0.670** | 0.609 |
| standard#11 | 0.764 | 0.625 | **0.633** | 0.596 |
| standard#12 | 0.821 | 0.636 | **0.662** | 0.609 |
| standard#13 | 0.810 | 0.697 | **0.711** | 0.645 |
| standard#14 | 0.842 | 0.641 | **0.664** | 0.597 |
| standard#15 | 0.829 | 0.550 | 0.579 | **0.580** |
| standard#16 | 0.823 | 0.611 | **0.638** | 0.561 |
| standard#17 | 0.846 | 0.628 | **0.647** | 0.581 |
| standard#18 | 0.808 | 0.643 | **0.678** | 0.606 |
| standard#19 | 0.813 | 0.633 | **0.643** | 0.613 |
| model_agreeing#0 | 0.747 | 0.895 | **0.872** | 0.751 |
| model_agreeing#1 | 0.783 | 0.845 | **0.842** | 0.823 |
| model_agreeing#2 | 0.697 | 0.911 | **0.878** | 0.800 |
| model_agreeing#3 | 0.811 | 0.852 | **0.847** | 0.794 |
| model_agreeing#4 | 0.790 | 0.885 | **0.866** | 0.837 |
| model_agreeing#5 | 0.734 | 0.869 | **0.859** | 0.840 |
| model_agreeing#6 | 0.750 | 0.820 | **0.811** | 0.806 |
| model_agreeing#7 | 0.696 | 0.869 | **0.861** | 0.819 |
| model_agreeing#8 | 0.656 | 0.836 | **0.832** | 0.832 |
| model_agreeing#9 | 0.797 | 0.853 | **0.837** | 0.787 |
| unfair#0 | 0.862 | 0.784 | **0.781** | 0.636 |
| unfair#1 | 0.843 | 0.795 | **0.795** | 0.766 |
| unfair#2 | 0.842 | 0.821 | **0.806** | 0.642 |
| unfair#3 | 0.841 | 0.799 | **0.786** | 0.644 |
| unfair#4 | 0.818 | 0.836 | **0.818** | 0.686 |
| unfair#5 | 0.833 | 0.830 | **0.816** | 0.706 |
| unfair#6 | 0.839 | 0.838 | **0.818** | 0.661 |
| unfair#7 | 0.807 | 0.819 | **0.809** | 0.747 |
| unfair#8 | 0.851 | 0.807 | **0.795** | 0.666 |
| unfair#9 | 0.849 | 0.782 | **0.781** | 0.665 |
| sparse#0 | 0.817 | 0.654 | **0.680** | 0.636 |
| sparse#1 | 0.839 | 0.706 | **0.758** | 0.700 |
| sparse#2 | 0.822 | 0.702 | **0.727** | 0.634 |
| sparse#3 | 0.803 | 0.673 | **0.684** | 0.611 |
| sparse#4 | 0.821 | 0.620 | **0.628** | 0.618 |
| sparse#5 | 0.813 | 0.651 | **0.680** | 0.601 |
| sparse#6 | 0.848 | 0.658 | **0.673** | 0.626 |
| sparse#7 | 0.798 | 0.679 | **0.689** | 0.620 |
| sparse#8 | 0.827 | 0.659 | **0.693** | 0.619 |
| sparse#9 | 0.826 | 0.717 | **0.732** | 0.660 |

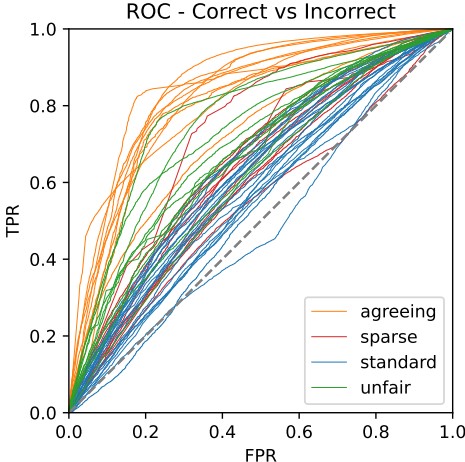

Figure 13: ROC-Curve - OvA Classifiers - Seed 1 - Test

We can see that on the vast majority of experts, the human expertise models are better at ranking the probability of error than the ova classifiers for each individual expert. This may be due to the fact that our model shares information across experts decisions, resulting in a better model of human behavior.

To evaluate the ranking across experts, we can evaluate the accuracy of the expert selected by each model, for each instance in the test set. For our human expertise model, this corresponds to the expert with the lowest associated cost, while for the OvA classifiers, this corresponds to the highest scoring classifier prediction. We then verify, for each instance, if said expert is correct. Since the misclassification cost is separate for label positive and label negative instances, we also calculate the accuracy within each of these subgroups.

Table 10: Top Expert Accuracy - averaged across seeds

| Method | All instances | Label positives | Label Negatives |
|--------|---------------|-----------------|-----------------|
| Hem    | 0.973         | 0.626           | 0.978           |
| OvA    | 0.975         | 0.526           | 0.982           |

As we can see the OvA classifiers result in a better ranking on the majority of cases. However, this is due to the fact that these classifiers are better at comparing expert correctness among label negative instances, while being significantly worse than our Human Expertise Model on the label positive instances, for which the misclassification cost is higher. We can then conclude that this novel architecture can lead to performance gains in high class imbalance, cost-sensitive scenarios.

