# OpenReview forum: "DeCCaF: Deferral Under Cost and Capacity Constraints Framework"
_ICLR.cc/2024/Conference — ICLR 2024 Conference Withdrawn Submission_

### Official Review · Reviewer_eN7q · 2023-10-30

**Soundness:** 2 fair
**Presentation:** 2 fair
**Contribution:** 2 fair
**Rating:** 3
**Confidence:** 3

**Summary:**

This paper studies the learning to defer (L2D) framework with some additional considerations: 1) neglecting cost-sensitive scenarios, 2) requiring concurrent human predictions for every instance of the dataset in training, and 3) not dealing with human capacity constraints. To address this problem, this paper proposes the deferral under the cost and capacity constraint framework. A novel L2D approach is proposed, which employs supervised learning to model the probability of human error with less restrictive data requirements (only one expert prediction per instance), and uses constraint programming to globally minimize error cost subject to capacity constraints. Experiments are conducted to validate the effectiveness of the proposed method.

**Strengths:**

- It is interesting to investigate L2D under the three additional considerations: 1) neglecting cost-sensitive scenarios, 2) requiring concurrent human predictions for every instance of the dataset in training, and 3) not dealing with human capacity constraints.
- A reasonable method is proposed to train the model for this new problem setting.

**Weaknesses:**

- My biggest concern lies in that the proposed method in this paper is completely heuristic. There are no theoretical analyses to theoretically justify the good properties of the proposed method. It is noteworthy that nearly all the previous works on L2D have provided theoretical guarantees for the proposed method. So I consider that the lack of theoretical guarantees would be a fatal drawback of this paper.
- Although the proposed method is reasonable, the novelty is quite limited. I cannot get any interesting insights by inspecting Eqs (1)-(3).
- Experimental results are insufficient and thus not convincing. More kinds of datasets with larger sizes should be used.

**Questions:**

Can existing methods be adapted to this problem setting?
Can more widely used benchmark datasets (i..e, CIFAR-10 and CIFAR-100) can be modified so that they can be used in this problem setting?

---

### Official Review · Reviewer_2idL · 2023-11-01

**Soundness:** 2 fair
**Presentation:** 2 fair
**Contribution:** 2 fair
**Rating:** 3
**Confidence:** 3

**Summary:**

This paper presents DeCCaF, a method that learns to defer to a human expert, with misclassification costs and capacity constraints in mind.

**Strengths:**

* Sections 1-3 are well-written.
* Weaknesses of prior algorithms are well-exposed in the introduction.

**Weaknesses:**

* In Section 4.1, there does not seem to be any motivation for the form of the equations used to describe the expert error probabilities.
* Table 1 compares the misclassification cost and predictive equality of the various approaches. Is there a reason you did not compare the recalls directly? Also, I'm confused why the predictive equality scores do not have error bars.
* The paper quickly gets confusing in Section 4. An overall algorithm describing the flow would help. For example, it isn't immediately clear that how the equations of Section 4.1 are related to the minimization objective of the previous section.
* Do you have results for each of the types of experts described in Section 4.1?
* I'm unclear where the 50 experts mentioned in the introduction are in the rest of the paper. Are those the simulated human experts?

**Questions:**

* Since $m_{i, j}$ is the label given to sample $i$ from expert $j$, does that mean this approach also needs all samples labeled by all experts?
* In Equation 3, when you instead have $\sum\limits_i A_{ij} \leq H_{b, j}$, does the performance change?

---

### Official Review · Reviewer_LUVz · 2023-11-01

**Soundness:** 2 fair
**Presentation:** 2 fair
**Contribution:** 1 poor
**Rating:** 3
**Confidence:** 4

**Summary:**

The authors consider the problem of learning to defer (L2D) assuming a team of experts.  More specifically, they focus on the problem of which expert should predict an instance, assuming that this was deferred to the experts.  For the team of human experts, they make the two following assumptions: a)  each expert has some limit on how many predictions they can make (capacity), and b) each expert incurs a (different) misclassification cost for each data sample, where the cost depends on their expected FP and FN rate. Under these assumptions, the authors propose training of a classifier  to predict for each instance and each expert if the expert prediction is a FP, FN, TP or TN. Then they use this model to  assign instances to the experts with minimum misclassification cost, while satisfying each expert’s capacity constraint. The authors evaluate their assignment method on a simulation scenario on fraud detection and show that their method outperforms competitive baselines.

**Strengths:**

The idea of assigning instances to experts, while taking into account their capacity and correctness is quite interesting and could have significant applications.

The paper is well organized, clearly structured and easy to read.

The experimental protocol is described in detail and the results appear reproducible.

**Weaknesses:**

Throughout the manuscript, the authors refer to their method as a learning to defer method. However, it seems that the main contribution of the current work is the method to assign instances to human experts, to minimize their classification error, assuming constraints on how many predictions each expert can make. The minimization objective in (3) refers to the problem of assigning instances to a team of experts, as it does not include the classifier. Learning to defer between experts and classifiers seems out of the main scope of the work, since the proposed method  does not assign the instance either to a classifier or a human expert (or team of experts). It would be less confusing, if it was clear that the proposed method addressed the problem of assignment instances to experts, assuming that a deferral policy selected the experts to predict.

**Novelty/Significance**

The proposed idea about a human expertise model does not appear as particularly novel. The vast literature of Discrete Choice Models has been about modeling the mechanism that individuals choose one option among a set of different options [5] and could be used in the context of classification on modeling human predictions. Moreover, modeling human expertise has been studied in cognitive psychology [3]. Particularly related to the L2D framework appears also [4], that shares the same motivation to reduce the need for human predictions and does opt for an ML model to simulate the experts predictions.  What could have strengthen the contribution would be to compare the proposed model with other expert models and show how/why this model is of greater benefit than others.

The main motivation of the proposed human expertise model is the scarcity of expert predictions. However, it is not clear how the proposed model  uses less data or why it (guarantees to) requires less data. One would expect that training an ML model to predict FP, FN, TP or TN for each expert and data sample would require a sufficient amount of data. A formal proof or experimental analysis on the data requirements of the proposed model and comparison with other models could provide evidence for the benefits of the proposed approach.

The problem of assigning instances to human experts seems quit relevant to the matching problem [1], and it would be helpful to clarify if algorithms on the matching problem can (or cannot) be used for this problem. If they are applicable, it would be necessary to compare them with the proposed method with experiments.

Accounting for capacity constraints on behalf of the experts is quite important, though it is not clear if/how these constraints cannot be captured/are different from the deferral cost in [6] or the triage level constraint in [7].

**Presentation**

It is confusing to use $m$ both for the model prediction and the expert prediction. Moreover, it is confusing that the authors use both $m$ and $m(x)$ for the ML model score.  Also, $t$ in section 4.1 is not defined.

Eq (3) is the constrained minimization objective to satisfy an assignment to human experts that minimizes the FPs and FNs under the capacity constraints. It is confusing to name the entire section (3.4) Minimization algorithm, the authors do not introduce an algorithm, they introduce the objective under constraints and then state that they use some off-the shelf technique (CP-SAT solver) to solve it.

The experimental results appear limited to the specific scenario. Since the authors perform simulations, it is not clear why they did not test their model using other datasets, such as the galaxy zoo dataset [8], and show that their proposed expert model outperforms baselines in other scenarios as well.

In terms of evaluation, since the proposed method focuses on the assignment of instances to experts, and not  a policy that selects either a classifier or a human (or team of humans) to make a prediction, comparison to L2D methods is not very well defined. Perhaps, it would be more  interesting to compare with algorithms within the literature of the matching problem [1] or bandits [2].

[1] Gerards, A. M. H. (1995). Matching. Handbooks in operations research and management science, 7, 135-224.

[2] Tran-Thanh, L., Stein, S., Rogers, A., & Jennings, N. R. (2014). Efficient crowdsourcing of unknown experts using bounded multi-armed bandits. Artificial Intelligence, 214, 89-111.

[3] Cooke, N. J. (2014). Modeling human expertise in expert systems. In The psychology of expertise (pp. 29-60). Psychology Press.

[4] Hemmer, P., Thede, L., Vössing, M., Jakubik, J., & Kühl, N. (2023). Learning to Defer with Limited Expert Predictions. arXiv preprint arXiv:2304.07306.

[5] Hensher, D. A., & Johnson, L. W. (2018). Applied discrete-choice modelling. Routledge.

[6] Mozannar, H., & Sontag, D. (2020, November). Consistent estimators for learning to defer to an expert. In International Conference on Machine Learning (pp. 7076-7087). PMLR.

[7] Okati, N., De, A., & Rodriguez, M. (2021). Differentiable learning under triage. Advances in Neural Information Processing Systems, 34, 9140-9151.

[8] Fortson, L., Masters, K., Nichol, R., Edmondson, E. M., Lintott, C., Raddick, J., & Wallin, J. (2012). Galaxy zoo. Advances in machine learning and data mining for astronomy, 2012, 213-236.

**Questions:**

1. In the introduction, it is not very clear what cost-sensitive means. Perhaps some clarification that by cost, the authors refer to the misclassification error (if so?), and explain that by cost-sensitive scenarios the authors refer to cases in which one requires guarantees on the classification error.
2. How do you account for the robustness of the human expertise model? What are the basic assumptions of the human expertise model?Does the human expertise model assume that the experts decide independently among each other?
3. How the “benchmark of synthetic expert decisions” can be applicable/useful in future works? As presented in section 1 it looks just as part of the evaluation on simulated  scenarios, but it is not clear how this benchmark is part of the contribution. Is this benchmark the “human behavior model” the authors use for their proposed DeCCaF framework?. Perhaps it would be helpful to first refer to the model and then to the DeCCaF framework.
4. Why do the authors do not use galaxy zoo or other datasets that do include expert predictions from multiple experts?
5. Under the paragraph “ML model” the authors refer to calibration of the output of the classifier. A formal definition for calibration would be useful to clarify what the authors mean by calibration.
6. How could the method be extended for multi class classification tasks?
7. The human expertise model  is trained to predict for each expert if their prediction is FP, FN, TP, TN. The method proposed in (2) and (3) that uses the human expertise model to decide which expert should predict the instance uses only the predictions on FP and FN. Since the output of the human expertise model is not used for TP and TN, could one merge these two classes and let $\mathcal{O} = \\{ FP, FN, T\\}$ for simplicity. Could this merge lead to a model that requires less data for training, as now we have 3 classes instead of 4? Is there a reason why $\mathcal{O}$ would be a better option?

---

### Official Review · Reviewer_Ygwy · 2023-11-05

**Soundness:** 2 fair
**Presentation:** 2 fair
**Contribution:** 2 fair
**Rating:** 5
**Confidence:** 4

**Summary:**

The paper proposed deferral mechanisms with cost and budget considerations built into them. This is a more realistic setting for the practical usability of the learning to defer systems. The experimental evaluation is done on a fraud-detection setting and empirical results are reported against two baselines.

**Strengths:**

1. The paper incorporate practical considerations into the learning to defer framework which is crucial for wider adoption of such a framework.
2. The experimental setup of the paper is realistic, and improves upon the simplistic setup of the related works in the literature.

**Weaknesses:**

1. The paper uses only OvA surrogate and regular rejection learning method as baselines. There are potentially other methods the authors can compare to (or comment on). For example, Hemmer et al. (2023) use model based imputation to deal with missing expert predictions (one of the major limitations of current L2D systems the paper highlights is that the current systems require expert predictions for all samples), or other baselines of incorporating costs and budget constraints directly into the existing L2D methods (see also question 1 below).

2. While that is not the focus of this paper, contemporary works in L2D have focused on theoretically sound algorithms as Mozannar and Sontag (2020) highlights the importance of them to avoid the sub-optimal allocations, the current paper does not comment on such considerations, and the method it employs is also a two-stage mechanism where such sub-optimality could be prevalent. It would be interesting to also comment on that.

Overall, while the paper does make good contributions for more practical usability of the L2D frameworks, the comparison with existing methods is lacking (experimentally as well as theoretically). Furthermore, more empirical evidence would be good to understand the efficacy of the proposed method.

Some minor weaknesses:
1. The notation of the paper can be heavily improved. Section 3.2: $j$ and $i$ both runs over $\{1, 2, \ldots, N\}$ which is confusing. Section 3.2 (Human expertise model): It is not specified what is $g(\cdot)$.

Hemmer et al. Learning to Defer with Limited Expert Predictions" (AAAI 2023)

**Questions:**

1. One major contribution of the paper is to incorporate instance dependent costs into the system. This could also be done based on the works of Cao et al. (2022) which eventually can result in theoretically grounded consistent surrogate optimisation objective. No such thing can be said for the proposed optimisation objective (as I understand). I'd be curious to hear from authors about this.

2. This could be misunderstanding on my part, but I understand that the the setting is that not all experts can provide the predictions for all the samples. Thus, in $S = \\{x_i, y_i, c_i, m_{i,j}\\}$, $m_{i,j}$ is assumed to be available for only one $j \in [J]$, or multiple experts are allowed to provide predictions for the same sample? In the latter case, how are missing predictions dealt with?



Cao et al. Generalizing Consistent Multi-Class Classification with Rejection to be Compatible with Arbitrary Losses. (NeurIPS 2022)